# Smoother versus sharper Gulf Stream and Kuroshio SST fronts: Effects on cyclones and climatology

Leonidas Tsopouridis[1], Thomas Spengler[1], and Clemens Spensberger[1]

[1]Geophysical Institute, University of Bergen, and Bjerknes Centre for Climate Research, Bergen, Norway

**Correspondence:** leonidas.tsopouridis@uib.no

**Abstract.** The Gulf Stream and Kuroshio regions feature strong sea surface temperature (SST) gradients that influence cyclone development and the storm track. Previous studies showed that smoothing the SSTs in either the North Atlantic or North Pacific yields a reduction in cyclone activity, surface heat fluxes, and precipitation, as well as a southward shift of the storm track and the upper-level jet. To what extent these changes are attributable to changes in individual cyclone behaviour, however, remains unclear. Comparing simulations with realistic and smoothed SSTs in the atmospheric general circulation model AFES, we find that the intensification of individual cyclones in the Gulf Stream or Kuroshio region is only marginally affected by reducing the SST gradient. In contrast, we observe considerable changes in the climatological mean state as well as a reduced cyclone activity in the North Atlantic and North Pacific storm tracks that are shifted equator-ward in both basins. The upper-level jet in the Atlantic also shifts equator-ward, while the jet in the Pacific strengthens in its climatological position and extends further east. Surface heat fluxes, specific humidity, and precipitation also respond strongly to the smoothing of the SST, with a considerable decrease of their mean values on the warm side of the SST front. This decrease is more pronounced in the Gulf Stream than in the Kuroshio region, due to the amplified decrease in SST along the Gulf Stream SST front. Considering the differences of the different variables occurring within/outside of a 750 km-radius of any cyclone over their entire lifetime, we find that cyclones play only a secondary role in explaining the mean state differences between the smoothed and realistic SST experiments.

## 1 Introduction

The Gulf Stream and Kuroshio regions with their strong sea surface temperature (SST) gradients are preferential locations for cyclogenesis (e.g., Hoskins and Hodges, 2002; Nakamura et al., 2004) and are found to determine the location and structure of storm tracks (e.g., Chen et al., 2010; Ogawa et al., 2012; Ma et al., 2015; Yao et al., 2018). Sensitivity tests with smoothed SSTs and a weaker SST gradient yield a reduced cyclone activity. In addition, these experiments feature an equator-ward shift of both the storm track as well as the upper level jet (e.g., Ma et al., 2015; Zhang et al., 2020) and a decrease of surface heat fluxes as well as precipitation on the warm side of the SST front (e.g., Kuwano-Yoshida et al., 2010b; Kuwano-Yoshida and Minobe, 2017). However, as it remains unclear if the latter changes can be attributed to changes in cyclone characteristics and activity, we quantify differences in cyclone behaviour and the contribution of cyclones to the documented differences when SSTs are changed in the Gulf Stream and Kuroshio regions.

SST gradients influence individual cyclone intensification (e.g., Sanders, 1986; Wang and Rogers, 2001; Jacobs et al., 2008), where the intensification has been associated with low-level baroclinicity originating from sensible heat fluxes (e.g., Hotta and Nakamura, 2011) and latent heating (e.g., Papritz and Spengler, 2015) along the SST front. However, other studies related the intensification of individual cyclones in the western Atlantic to the low-level baroclinicity associated with the pronounced

land-sea contrast (e.g., Brayshaw et al., 2009; Tsopouridis et al., 2020a). Thus, while several studies highlighted the sensitivity of cyclogenesis and the storm track to a smoothing of the SST (e.g., Nakamura et al., 2008; Kuwano-Yoshida and Minobe, 2017; Ma et al., 2017; Zhang et al., 2020), the impact of a weaker SST gradient on the intensification of individual cyclones remains unclear.

Randomly selecting 24 individual cyclones that occurred in the Gulf Stream region, de Vries et al. (2019) highlighted the
reduction of surface latent heat fluxes and low-level baroclinicity when smoothing the SST. They, however, emphasised that these changes vary based on the position of each storm relative to the SST front. Similarly, Tsopouridis et al. (2020a) found cyclones following different pathways with respect to the SST front position to be associated with different characteristics. They, however, attributed the structural changes primarily to the absolute SST over which the cyclone is propagating rather than the SST front itself. This is consistent with the idealised simulations of cyclone development using different SST and SST
gradients by Bui and Spengler (2021), who identified a primary dependence on cyclone development to absolute SSTs and only minor dependence on the SST gradient. Similarly, Booth et al. (2012) found strength of storms to increase in the Gulf Stream region with increased SSTs, even if only a weak SST gradient is present. Overall, the twofold dependence on both the absolute SST and the strength of the SST front indicates that both influence the development of cyclones.

In addition to low-level baroclinicity, upper-level forcing by the jet stream is known to contribute to cyclogenesis (e.g.,
Sanders and Gyakum, 1980; Uccellini et al., 1984; Sinclair and Revell, 2000; Yoshida and Asuma, 2004) and can influence cyclone intensification (e.g., Evans et al., 1994; Schultz et al., 1998; Riviere and Joly, 2006; Tsopouridis et al., 2020b). At the same time, the very existence of the extratropical jet depends on cyclones maintaining the storm track (Hoskins and Valdes, 1990; Holton and Hakim, 2012; Papritz and Spengler, 2015). Accordingly, using experiments with realistic and smoothed SSTs, Kuwano-Yoshida and Minobe (2017) argued that the increased cyclone activity over the SST front influences the upper-level
jet, causing its meandering over the North Pacific.

In the light of this tight coupling between the jet and the storm track, it is not surprising that a smoothing of the SST can affect the upper-level flow. Indeed both the storm track (e.g., Small et al., 2014; Ma et al., 2015; Piazza et al., 2016; Zhang et al., 2020) and the upper-level jet (e.g., Ma et al., 2017; O'Reilly et al., 2017) were shown to shift equatorward in the North Atlantic and Pacific ocean when the SSTs were smoothed. Kuwano-Yoshida and Minobe (2017) showed that smoother SST
in the Kuroshio region resulted in a more zonally oriented storm track and argued that a weaker SST front is not able to anchor the upper-level flow. A southward shift of both the storm track density and the upper-level jet when smoothing the SST has also been documented in the North Atlantic region by Piazza et al. (2016), though their shift of the storm track was smaller compared to the one in the study by Small et al. (2014), which they related to the stronger SST smoothing. Based on the aforementioned arguments, a smoothing of an already climatologically weaker SST front in the Kuroshio region (e.g.,
Nakamura et al., 2004; Tsopouridis et al., 2020b) should have a comparatively minor impact on the storm track and the upper-

level wind speed compared to the Gulf Stream region. Thus, even while some recent studies emphasise the impact of mesoscale eddies (e.g., Bishop et al., 2017; Liu et al., 2021), it is important to further understand the influence of these larger scale SST gradients.

Focusing on mesoscale aspects of the atmospheric response to a smoothing of the SSTs, Piazza et al. (2016) documented a considerable decrease in the surface heat fluxes (30-50%) and convective precipitation (up to 60%) over the warm side of the North-Atlantic SST front after they removed small-scale SST features. Consistently, Zhang et al. (2020) found a similar, yet significantly smaller, decrease of the surface heat fluxes (5%) and precipitation (7%) within the Kuroshio and Oyashio confluence region. Atmospheric general circulation model (AGCM) experiments with real and smoothed SSTs revealed that the SST front is important to maintain convective precipitation (in line with Minobe et al., 2008), with the atmospheric response of the SST smoothing being stronger in the Gulf Stream than in the Kuroshio region (Kuwano-Yoshida et al., 2010b). Indeed, comparing differences in precipitation between the original and smoothed SSTs as well as between the Atlantic (Minobe et al., 2008) and the Pacific (Kuwano-Yoshida and Minobe, 2017), the decrease of precipitation is more pronounced in the Gulf Stream region. Thus, surface heat fluxes and precipitation are considerably affected by the strength of the SST gradient, with the effect being stronger in the Gulf Stream than in the Kuroshio region. However, whether the time-mean distributions of such atmospheric patterns are only altered by the SST gradients or to what extent changes in the occurrence or intensification of cyclones contribute to their distribution remains ambiguous.

While the spatial distribution of surface wind convergence into a narrow band has been linked to strong SST gradients (Small et al., 2008), recent studies associated the mean state's characteristics in the Gulf Stream and Kuroshio regions to synoptic features (e.g., O'Neill et al., 2017; Parfitt and Seo, 2018). In particular, O'Neill et al. (2017), associated the existence of the Gulf Stream convergence zone with intense cyclones propagating in the region and highlighted the overall role of storms in shaping the mean state of the atmosphere in the northwest Atlantic. More specifically, Parfitt and Seo (2018) pointed out the importance of atmospheric fronts for the climatological near-surface wind convergence over the two regions. Masunaga et al. (2020a, b), on the other hand, showed that strong cyclones and atmospheric fronts only have a minor contribution to the climatological mean convergence in the Gulf Stream and Kuroshio regions and that the main contribution is associated with weaker storms and fronts. However, given the weakness of these systems, it could be questioned how significantly the climatological contribution is associated with fronts and storms in general. In fact, Reeder et al. (2021) proved that the direct impact of SST fronts on atmospheric fronts is negligible, which they confirmed by a climatological analysis for the Atlantic.

Extratropical cyclones strongly modulate the horizontal moisture transport (e.g., Ruprecht et al., 2002; Chang and Song, 2006) and precipitation (e.g., Bjerknes, 1922; Pfahl and Wernli, 2012; Hawcroft et al., 2012). While surface heat fluxes can have a direct and indirect effect on cyclone development (e.g., Haualand and Spengler, 2020), the role of cyclones shaping heat fluxes is under debate. Some studies suggest a close relationship between surface heat fluxes and cyclones in the midlatitudes on both synoptic (e.g., Alexander and Scott, 1997; Schemm et al., 2015; Dacre et al., 2020) and longer time scales (e.g., Parfitt et al., 2016; Ogawa and Spengler, 2019). Using a more statistical approach, Zolina and Gulev (2003) argued that the surface fluxes mainly occur on synoptic time scales. However, based on a composite analysis Rudeva and Gulev (2011) indicated that cyclones in the North Atlantic do not contribute significantly to the climatological surface heat fluxes in this region.

Furthermore, Tanimoto et al. (2003) noted that in regions with active ocean dynamics, such as along the western boundary currents, the SST anomalies mainly regulate the surface heat fluxes and not the cyclones.

To shed light on these aforementioned issues, we assess the effect of a weak or strong SST gradient using an atmospheric general circulation model (AFES 3) based on simulations with realistic and smoothed SSTs in the Gulf Stream and Kuroshio regions. Our analysis of these simulations is twofold. Firstly, we follow the approach of Tsopouridis et al. (2020a, b, hereafter TSSa,TSSb) with the aim to quantify the effect of the smoothed SSTs on the structure and characteristics of individual cyclones. Secondly, to assess the climatological role of cyclones to changes in the magnitude of the SST front, we consider pertinent variables within and outside a radius around cyclone centres in the Atlantic and Pacific basin throughout their lifetime to examine the contributions of cyclones to the wintertime climatology for the realistic and smoothed simulations (similar to Ma et al., 2015). This two-pronged approach allows us to establish a connection between structural changes in individual cyclones and changes in the time-mean winter climatology.

## 2 Data and Methods

### 2.1 Data

We use data from version 3 of the AGCM for the Earth Simulator (AFES) developed by the Earth Simulator Center of the Japan Agency for Marine Earth Science and Technology (JAMSTEC, Ohfuchi et al., 2004; Enomoto et al., 2008; Kuwano-Yoshida et al., 2010a). This version of AFES has been first used by Kuwano-Yoshida and Minobe (2017) and O'Reilly et al. (2017) and has a horizontal resolution of T239 (approximately 50 km) and 48 sigma levels in the vertical. The model was integrated from 1 September 1981 to 31 August 2001, where we only focus on the winter periods December to February, hereafter DJF. Throughout the time period, NOAA 0.25° daily SST data (Reynolds et al., 2007) were used as boundary conditions. For our analysis we use the AFES output on a 0.5° horizontal grid at 6 hourly intervals. More information about the model configuration can be found in Kuwano-Yoshida and Minobe (2017).

Using AFES 3, Kuwano-Yoshida and Minobe (2017), produced two experiments for the North Pacific. Firstly, the control experiment (hereafter CNTL) that uses the original global SST data and secondly an experiment that uses smoothed SSTs over the greater area around the Kuroshio Extension (hereafter SMTHK). They also composed an analogous experiment with spatially smoothed SSTs over the greater area around the Gulf Stream (hereafter SMTHG). In both cases, the NOAA 0.25° daily SST were smoothed by applying a 1-2-1 running mean filter 200 times in the zonal and meridional direction. It is a three-point filter with the weights 0.25, 0.5 and 0.25, that has a sharp cutoff frequency, so that unwanted frequency components are effectively removed.

We use SST, latent and sensible heat fluxes, large-scale and convective precipitation, specific humidity at 850 hPa, and wind at 300 hPa for our analysis. We also compare the model simulations with the same variables from the ERA-Interim reanalysis that was created using a four-dimensional variational data assimilation scheme and a spectral truncation of T255 and 60 levels in the vertical (Dee et al., 2011).

## 2.2 SST front and jet stream detection

We identify the position of SST fronts using an objective frontal detection scheme based on the "thermal" method, as described
in detail by TSSa. This method has also been used to detect atmospheric fronts (Jenkner et al., 2010; Berry et al., 2011; Schemm
et al., 2015). We perform the detection using SST data filtered with a spectral truncation to T84 resolution and detect the SST
fronts in the instantaneous SST field every 6-hours. We detect SST front lines to define when cyclones cross the front. After
thoroughly testing different thresholds for the two regions, we use a threshold of 2K/100km for the Atlantic and a smaller
threshold of 1.25K/100km for the Pacific region to capture the most prominent SST front lines in the respective regions. The
use of two different thresholds is necessary due to the different strength of SST gradients in the two regions and thus to ensure
that an SST front along the Kuroshio is detected sufficiently regularly while at the same time avoiding the detection of spurious
SST gradients along the Gulf Stream. This choice is in line with the different characteristics of the two boundary currents
described in (Nakamura et al., 2004) and the thresholds used in previous studies for the Atlantic (TSSa) and Pacific (TSSb).

To assess the potential impact of the SST smoothing on the upper levels, we detect the position of jet following the algorithm
of Spensberger et al. (2017). The algorithm detects jet axes, lines of maximum wind that separate cyclonic and anticyclonic
wind shear. The algorithm requires the wind maximum to be well-defined, but does not impose a strict minimum wind speed
(Spensberger et al., 2017).

For the climatologies and composites we normalise the occurrence of both SST front lines and jet axis lines to account for
the latitudinally varying area covered by grid cells. We achieve this by showing the average length of SST front line/jet axis
line per unit area, hence the resulting unit of length per area. For details on the normalisation we refer to the jet climatology by
Spensberger and Spengler (2020).

## 2.3 Cyclone detection and tracking

We employ the University of Melbourne cyclone detection and tracking algorithm (Murray and Simmonds, 1991a, b). The
algorithm detects maxima in the Laplacian of the sea-level pressure field and tracks them over time using a nearest-neighbour
method together with the most likely direction of propagation (Murray and Simmonds, 1991a, b; Michel et al., 2018; TSSa;
TSSb). We use the same tracking and detection namelist as TSSa, in which the sensitivity of the results has been thoroughly
tested using different values for the selected parameters. We consider cyclone tracks with at least five 6-hour time steps in the
North Atlantic or the North Pacific and require the great circle distance between cyclogenesis and cyclolysis to be greater than
300 km in order to remove quasi-stationary systems.

The cyclone density pattern is in good spatial agreement with previous studies (Hanley and Caballero, 2012; Neu et al.,
2013; TSSa; TSSb), successfully capturing the major regions of cyclone activity in both basins. Consistent with the results of
TSSa and TSSb for ERA-Interim, we relate the small quantitative differences in the density climatology presented by both Neu
et al. (2013) or Murray and Simmonds (1991a), who also used the Melbourne University algorithm, to the neglect of shallow
and weak systems in our database.

## 2.4 Classification of cyclone tracks based on their position relative to the SST front

We categorise the identified cyclone tracks with a maximum intensification in either the Gulf Stream region (30-50°N and 290-310°E) or the Kuroshio region (30-50°N and 145-170°E) based on their propagation relative to the SST front. Analogous to TSSa and TSSb, for this classification we consider only cyclones with at least three 6-hour time steps in the Gulf Stream and Kuroshio regions. Firstly, we identify the shortest distance between each cyclone position and the SST front line for every time-step along the cyclone track and define a vector pointing to the cyclone. Then, we follow TSSa and TSSb and focus on cyclones that always stay on the cold (C1) or warm (C2) side of the SST front, and those that cross the SST front from the warm to the cold side (C3).

We categorise the cyclones for the SMTHG and SMTHK experiments analogously to the CNTL experiment. However, as the SST gradient in the smoothed experiments is very homogeneous over a large region and thus too weak to qualify as a front, we instead use the front positions from the CNTL experiment for the classification. We use the same classification as in CNTL to be able to compare cyclones with geographically similar genesis locations and tracks across the experiments. For simplicity we still refer to C1-3 as the cold, warm, and crossing cases for the smooth (SMTH) experiments, even though, strictly speaking, no SST front is crossed.

## 2.5 Decomposition of climatological differences

In addition to the cyclone track classification, we present a decomposition of the winter climatology for selected variables, where we conditioned the two composites on either occurring within or outside an area with a radius around a cyclone centre throughout the life cycle of a cyclone. We performed this analysis for each ocean basin, irrespective of the direction of cyclone propagation and location of its maximum intensification. Consistent with the threshold on cyclone circumference in Wernli and Schwierz (2006) and the analysis of Rudeva and Gulev (2011), we choose a radius of 750 km. We also obtained results for a radius of 500 and 1000 km, respectively (Fig. S5-S8). However, given that the results were not very sensitive to this choice, we focus on results with a radius of 750 km, for which most cyclone-related features, such as fronts, are included.

## 3 Results and Discussion

### 3.1 SST front and SST/SST gradient distribution between the experiments

Analysing the SST (Fig. 1a) and SST gradient (Fig. 2a) distribution in the Gulf Stream region (black box) for CNTL, we note a remarkable SST contrast across the Gulf Stream, which results in a strong SST gradient (Fig. 2a) and in locally well-confined SST front detections (Fig. 3a), consistent with an oceanographic viewpoint (Meinen and Luther, 2016). The SST front distribution also resembles the correspondent feature presented in TSSa for the same region, but based on a different period and dataset.

In the Kuroshio (black box in Fig. 1d), we observe a similar but spatially less confined SST contrast compared to the Gulf Stream region (compare Fig. 1a with Fig. 1d), which results in a weaker SST gradient in the Kuroshio region (Fig. 2d).

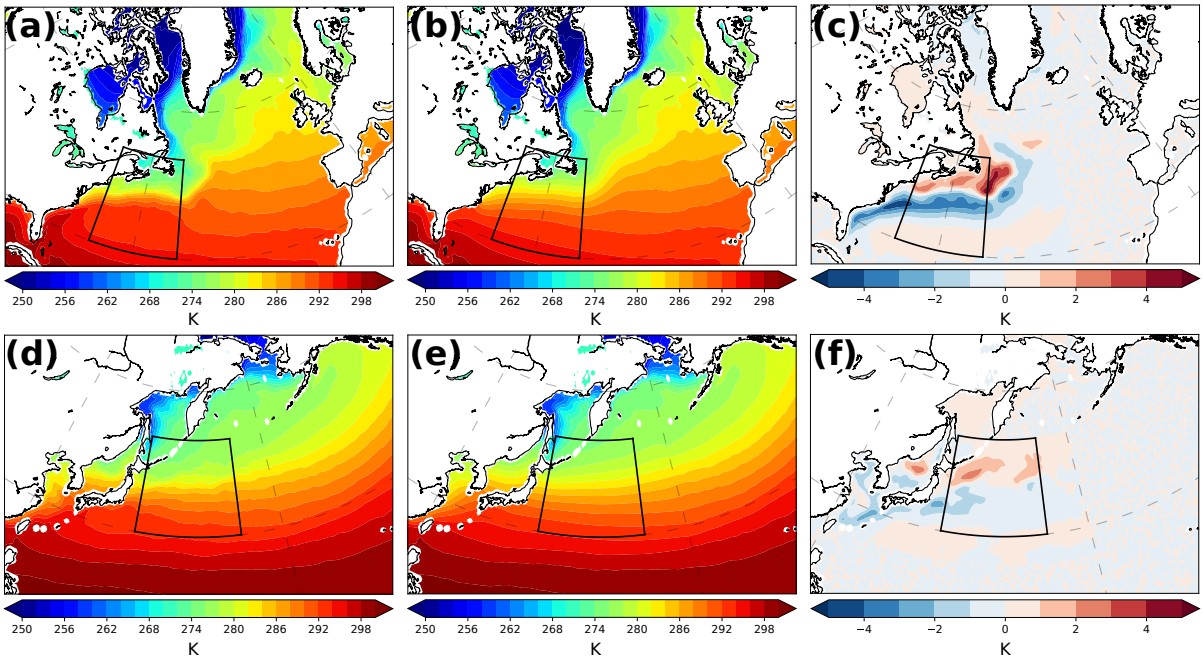

**Figure 1.** Climatological SST for DJF for (a) CNTL, (b) SMTHG, and (c) difference SMTHG-CNTL [K]. (d-f) As (a-c), but for the North Pacific. The Gulf Stream and Kuroshio regions are marked with a black box, respectively.

Consequently, the detected SST fronts are more wide-spread in the Pacific (Fig. 3b), but remain collocated with the region of the climatologically largest SST gradient (in line with, e.g., Tozuka et al., 2018; Wang et al., 2019). The less pronounced SST gradient in the Pacific compared to the Atlantic is also evident in the ERA-Interim winter climatology (Fig. S1 b,e), with the differences between the reanalysis and AFES simulations arising from the coarser SST resolution used in ERA-Interim prior 195 to 2002 (e.g., Masunaga et al., 2015; Parfitt et al., 2017).

Compared to CNTL, the SSTs in SMTHG are smoother and their gradient is more widely distributed (compare Fig. 1a,b and Fig. 2a,b). In the western Atlantic, we also observe that the smoothing affects SSTs at a considerable distance from the Gulf Stream SST front (Fig. 3a), for example reducing the SST to the east of the Florida Peninsula (Fig. 1a,b). At approximately 40°N, the SST differences exhibit a clear dipole, with increased SST to the north and decreased SST to the south, following the 200 position of the SST front (Fig. 1c, Fig. 3a). The largest differences occur offshore off the central US East coast ($\Delta$SST$< -4K$) as well as off the coast of Newfoundland ($\Delta$SST$> 4K$; Fig. 1c).

The SST distribution in SMTHK (Fig. 1e) is similar to CNTL, though smoother, and the region with the strongest gradients off the east coast of Japan is oriented slightly more zonally (Fig. 1d). Contrary to the Gulf Stream region, the SSTs outside the Kuroshio region remain largely unaffected. As in the Gulf Stream region, the differences in SST between the two experiments 205 follow a bipolar structure (Fig. 1f), but they are considerably weaker. The smoothing results in a maximum decrease (increase)

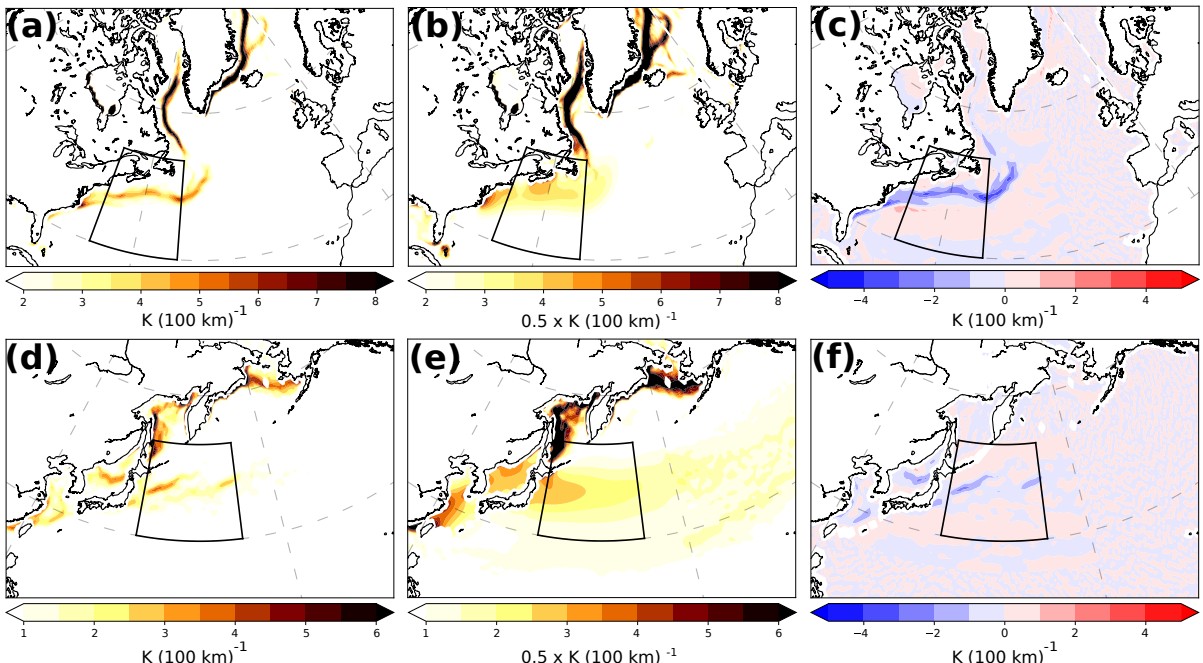

**Figure 2.** (a) Climatological SST gradient for DJF for (a) CNTL [K (100 km)$^{-1}$], (b) SMTHG [0.5 K (100 km)$^{-1}$], and (c) difference SMTHG-CNTL [K (100 km)$^{-1}$]. (d-f) As (a-c), but for the North Pacific. The Gulf Stream and Kuroshio regions are marked with a black box, respectively.

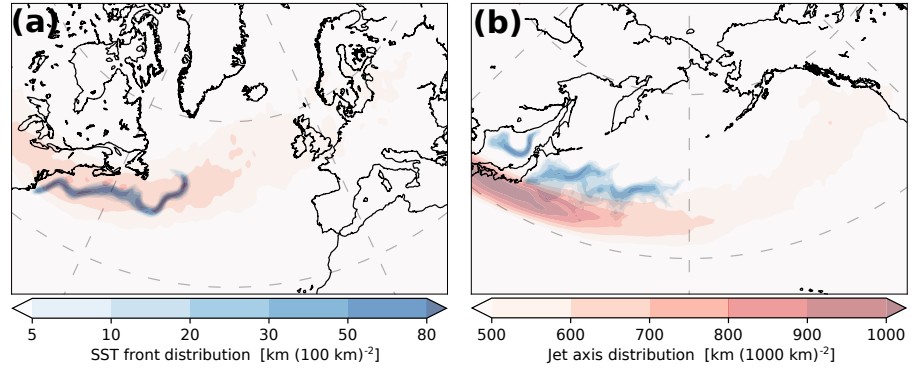

**Figure 3.** (a) SST front distribution (blue shading, km of line (100 km)$^{-2}$) and jet axis distribution (pale red shading, km of jet axis line (1000 km)$^{-2}$) for the North Atlantic. (b) As (a), but for the North Pacific

of 2 K (3 K) south (north) of the SST front (Fig. 1f), with the SST differences being more pronounced in the western part of the domain close to Japan (Fig. 1f).

## 3.2 Cyclone Density and Upper-level Wind

The position of the North Atlantic jet coincides with the location of the SST front (Fig. 3a), whereas the Pacific jet is located south of the Kuroshio SST front (Fig. 3b). The Pacific jet is stronger, meridionally more confined, and located over lower latitudes compared to the Atlantic jet, which is consistent with previous studies (e.g., Nakamura, 1992; Spensberger and Spengler, 2020). Both the strength and the position of the jet axes distribution in the North Atlantic are analogous to the study of TSSa, while for the Pacific we observe fewer jets in the region to the south of Japan compared to TSSb.

The changes in the SST field between CNTL and SMTHG/SMTHK introduce differences in both cyclone density and the climatological jet stream position. We observe an equatorward shift in the maximum cyclone density in both the North Atlantic and North Pacific, particularly in the central and eastern part of the basins (Fig. 4a,c). We also notice an analogous shift in the upper-level jet in the North Atlantic (Fig. 4b), with negative anomalies in the northern part of the basin and positive anomalies mainly to the east of the Gulf Stream region. However, a similar shift of the upper-level jet is not observed in the North Pacific, with the position of the jet remaining rather unchanged and the jet being more confined and zonally elongated in SMTHK (Fig. 4d).

While cyclone density (Fig. 4a) and jet occurrence (Fig. 4b) both shift equatorward in the Atlantic in the SMTHG experiment, there is no clear relation between these difference fields in the Pacific (Fig. 4c,d). For example, cyclone density in the Northeast Pacific (Gulf of Alaska, Fig. 4c) increases, whereas upper-level wind speed decreases in this region (Fig. 4d). Zhang et al. (2020) found a similar decrease in the upper-level response in the northeastern Pacific, expressed by differences of meridional eddy wind variance and eddy kinetic energy at 300 hPa. However, they showed that the upper-level decrease was not accompanied by a reciprocal negative anomaly in the low-level storm track and thus documented a different response of the SST smoothing in the lower and upper levels in the Pacific.

## 3.3 Classification and Intensification of Cyclones

To assess the effect of smoothing the SSTs on the evolution of individual cyclones, we now restrict our focus to cyclones with maximum intensification in the Gulf Stream or Kuroshio region (details in section 2.4). For CNTL in the Atlantic, 57 cyclones consistently stay on the cold side of the Gulf Stream SST front (C1), 27 cyclones stay on the warm side (C2), and 40 cyclones cross the SST front from the warm to the cold side (C3). In SMTHG, 62, 30, and 25 cyclones belong to C1, C2, and C3, respectively. Comparing these numbers, we notice that the number of cyclones staying on either side of the SST front is nearly unaffected by the smoothing, whereas the number of crossing cyclones is substantially reduced. This implies an overall reduction in the number of cyclones, which is in line with the decreased cyclone density along the Gulf Stream SST front in SMTHG (Fig. 3a, Fig. 4a).

In the Kuroshio region, the number of cyclones in C1 (86/81) and C3 (59/60) is more or less unchanged between CNTL/ SMTHK, whereas cyclones in C2 (24/14) decrease slightly in number. The small number of cyclones in C2, particularly in SMTHK, implies some uncertainty for the corresponding results. Note that in contrast to the Atlantic, there is no reduction in

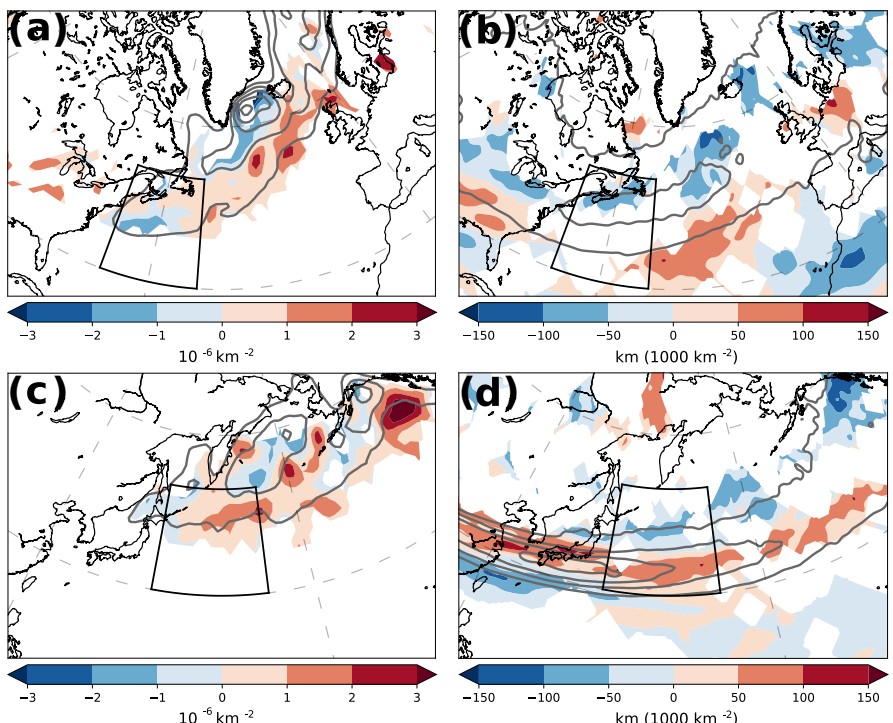

**Figure 4.** (a) Statistically significant (>95%, based on the chi-square test) difference in climatological cyclone density (SMTHG-CNTL, shading, $10^{-6}$ km$^{-2}$) and climatological cyclone density for CNTL (contours, first contour: $6*10^{-6}$ km$^{-2}$, interval $3*10^{-6}$ km$^{-2}$ ). (b) Statistically significant (>95%, based on the chi-square test) difference in climatological jet stream density (SMTHG-CNTL, shading, km of jet axis line $(1000 \text{ km})^{-2}$) and climatological jet stream density for CNTL (contours, first contour: 400 km of jet axis line $(1000 \text{ km})^{-2}$, interval: 200 km of jet axis line $(1000 \text{ km})^{-2}$. (c,d) As (a,b), but for the North Pacific.

240    cyclones crossing the SST front, potentially because the SST gradient in the Pacific is already comparatively weak in CNTL (compare Figs. 2a, d).

The more pronounced reduction of cyclones crossing the SST front in SMTHG compared to SMTHK highlights the significance of a strong SST gradient to anchor the position of the storm track, as discussed in previous studies (e.g., Nakamura et al., 2004, 2008; Sampe et al., 2010). Our results confirm these studies not only based on the number of cyclones, but also

245    based on the location of cyclones during their maximum intensification. In CNTL, most cyclones undergo their most rapid intensification close to the SST front (Fig. S3 b), whereas the location of most rapid intensification is distributed over a wider region in SMTHG (Fig. S3 e).

Among the three categories, Atlantic cyclones of C3 and C1 are deepening the most in CNTL, with a maximum 6-hour intensification corresponding to approximately 28 and 25 hPa/day, respectively. Conversely, cyclones of C2 are characterised

250    by a weaker intensification throughout their evolution (Fig. 5a). The results for C2 are statistically significant around the time of maximum intensification (from -6h to 6h), as the $50^{th}$ percentile of C2 coincides with the $75^{th}$ percentile of C1 and with the

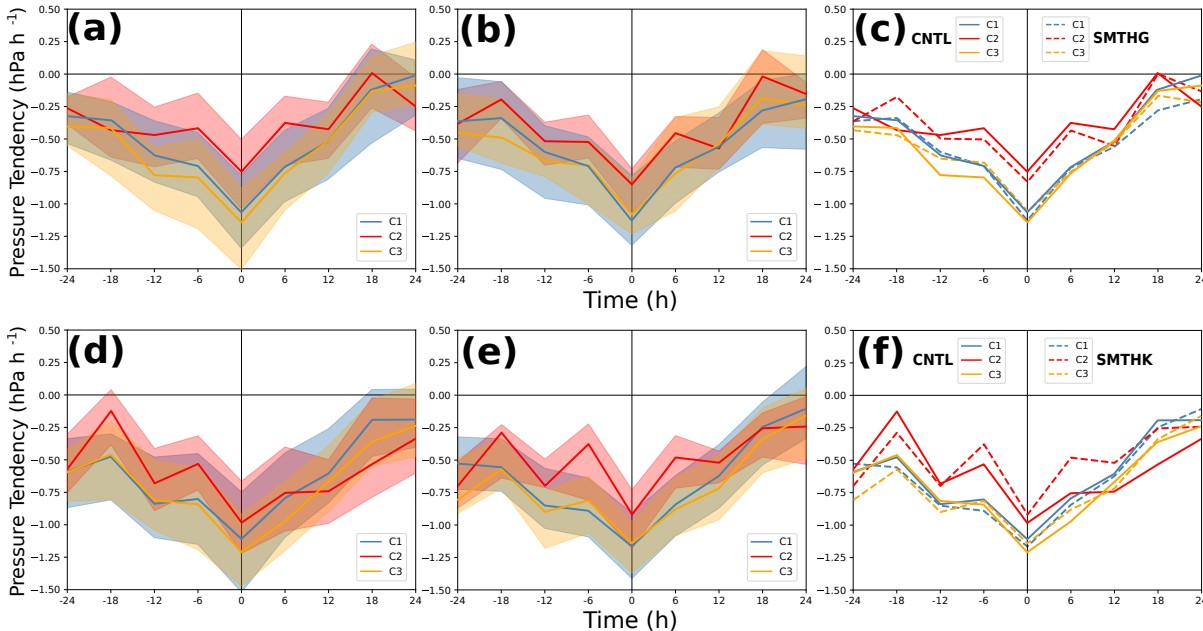

**Figure 5.** Pressure tendency (hPa h$^{-1}$) for the three categories relative to the time of maximum intensification for (a) CNTL and (b) SMTHG. Lines indicate the median and the shading the interquartile range. (c) Medians of the pressure tendency (hPa h$^{-1}$) for the three categories relative to the time of maximum intensification for CNTL (solid lines) and SMTHG (dashed lines). (d-f) As (a-c), but for the North Pacific.

75-100$^{th}$ percentile (not shown) of C3. Moreover, we notice that the median (50$^{th}$ percentile) of C2 is always above the ones for C1 and C3 during the 48 hour time period in both experiments for the Gulf Stream region, indicating a clear tendency for higher intensification in C1 and C3, compared to C2. In SMTHG, cyclones intensify similarly fast as in CNTL (Fig. 5a,b). In particular, cyclones of C3 experience only a slightly weaker intensification in SMTHG, although the SST front that they cross is barely existing in SMTHG.

These results support the findings of TSSa, who related the intensification of cyclones in the Gulf Stream region to the low-level baroclinicity associated with the land-sea contrast, hypothesising that a strong SST gradient only weakly modifies the deepening of the cyclones. To further clarify the potential role of the land-sea contrast and the SST front on cyclone intensification, we present cyclone-relative composites for the three categories. In CNTL, cyclones in C1 are associated with stronger low-level baroclinicity, i.e., a stronger temperature gradient at 850 hPa, compared to C2 (Fig. S11), because cyclones in C1 propagate close to the United States coast. Cyclones in C2, however, are characterised by a more maritime propagation (Fig. S9 c,e). Cyclones in C3 are associated with an equally strong temperature gradient at 850 hPa (Fig. S11 g-i) as in C1, even though cyclones propagate close to both the coastline and the SST front (Fig. S9g). In SMTHG, the location of cyclones is rather unchanged compared to CNTL (Fig. S9), but cyclones propagate over a considerably weaker SST front. However, for cyclones in C3, the temperature gradient at 850 is approximately the same as in CNTL (compare Figs. S11 g-i with S12 g-i), indicating the dominant role of the land-sea contrast to enhance low-level baroclinicity and hence cyclone intensification.

In the Kuroshio region, cyclones of C3 are deepening the fastest (approx. 30 hPa/day), followed by cyclones of C1 (approx. 25 hPa/day; Fig. 5d). There is a considerable overlap in the interquartile range for the 3 categories and a larger variability compared to the Gulf Stream region. Nevertheless, the median of C2 is also above the ones for C1 and C3 before the time of maximum intensification, indicating a clear tendency for weaker intensification in C2, compared to C1 and C3. In line with TSSb, C2 becomes the category with the larger deepening rate among the three categories after the time of maximum intensification. However, in contrast to TSSb, cyclones of C2 deepen the least before their maximum intensification in both CNTL and SMTHK (Fig. 5d,e). We relate this difference to cyclones in the AFES simulations being located further away from an upper-level jet stream than in the ERA-Interim data used in TSSb (not shown), where the upper levels appear to substantially influence the intensification of cyclones in the Kuroshio region. Moreover, the limited number of cyclones in the AFES simulations lowers the statistical robustness of these results (24 cyclones in CNTL compared with 97 cyclones in TSSb).

In SMTHK, the cyclones in the three categories have similar pressure tendencies as in CNTL (Fig. 5e), which becomes even more apparent when comparing the median of their pressure tendencies (Fig. 5f). In particular, C3 intensifies virtually identically in the two experiments. In the light of the corresponding findings for the Gulf Stream region, this result is not surprising. Even for the more focused Gulf Stream SST front, the smoothing had only a minor impact on the evolution of C3 cyclones (Fig. 5c,f). A more pronounced effect is evident for C2 cyclones in the Kuroshio region, with cyclones intensifying slightly less after the smoothing (Fig. 5f). However, the number of cyclones is even smaller in SMTHK (14 cyclones) than in CNTL, making it difficult to draw conclusions from this difference.

Considering the evolution of the SST underneath the cyclone core, cyclones of C1 propagate over comparatively low SSTs, because they remain on the cold side of the SST front in both regions (Fig. 6a,d). In contrast, cyclones of C2 propagate over approximately 16 K higher SSTs than cyclones of C1 in the North Atlantic (Fig. 6a) and over 12K higher SSTs in the North Pacific (Fig. 6d) during maximum intensification. However, during their evolution, cyclones gradually propagate towards lower SSTs. Cyclones of C3 propagate over higher SSTs at an early stage of their development and over lower SSTs after crossing the SST front. The latter is associated with the cross-frontal SST difference, which is more pronounced in the Atlantic (Fig. 6a) than in the Pacific (Fig. 6d) due to the sharper Gulf Stream SST front (consistent with Nakamura et al., 2004; Joyce et al., 2009; TSSb).

In the smooth experiments, cyclones of C1 propagate over approximately 1 K higher SSTs in both regions and cyclones of C2 over 1-2 K lower SSTs. Further, cyclones of C3 in the smoothed experiments experience a less sharp decrease in SST across the SST front compared to CNTL (Fig. 6b,c,e,f). The SST differences introduced by the smoothing are more pronounced before maximum intensification, because at this stage cyclones typically propagate in the western part of the regions of interest where the SST differences associated with the smoothing tend to be larger (Fig. 1c,f).

Overall, the considerable reduction in the number of cyclones of C3 after the smoothing of the SST in the Atlantic highlights the anchoring effect of a strong Gulf Stream SST front on the storm track. On the other hand, the already weak SST gradient in the Kuroshio prior to the smoothing leads to minor SST differences between CNTL and SMTHK and to a similar number of cyclones that cross the SST front. The rather similar cyclone intensification between the experiments indicates that the SST gradient is not particularly important for the intensification of individual cyclones (consistent with TSSa; TSSb).

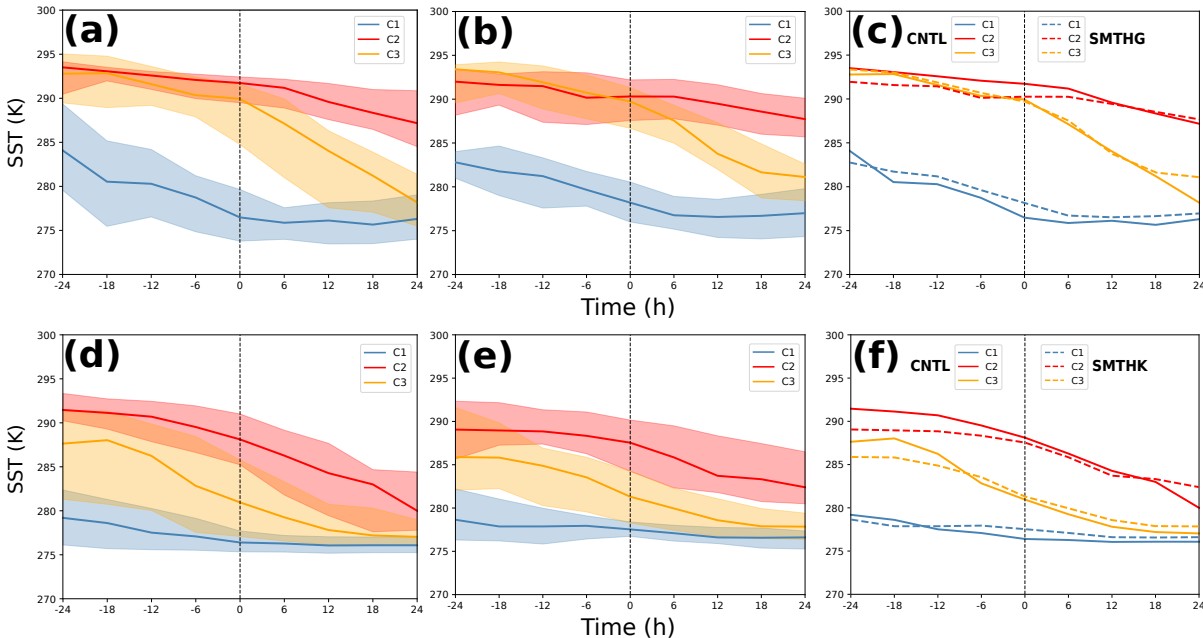

**Figure 6.** SSTs (K) for the three categories relative to the time of maximum intensification for (a) CNTL and (b) SMTHG. Lines indicate the median and the shading the interquartile range. (c) Medians of the SST [K] underneath the cyclone for the three categories relative to the time of maximum intensification for CNTL (solid lines) and SMTHG (dashed lines). (d-f) As (a-c), but for the North Pacific.

Our results thus suggest that the SST smoothing does not result in significant differences in the characteristics of individual cyclones, which is consistent with Bui and Spengler (2021), who found that cyclone development is more sensitive to absolute SST than SST gradients. This result is supported by cyclone-relative composites of, for example, low-level baroclinicity (Fig. S11,S12), surface heat fluxes, and precipitation (not shown), which exhibit only minor differences between the SMTH and CNTL experiments for all cyclone categories. Nevertheless, we observe a latitudinal shift in the storm track and the jet stream climatologies associated with the SST smoothing (Fig. 4) and previous studies documented a significant reduction in the climatological surface heat fluxes and precipitation (e.g., Kuwano-Yoshida et al., 2010b; Kuwano-Yoshida and Minobe, 2017). These discrepancies raise the question how the largely unaffected characteristics of individual cyclones relate to the evident changes in the climatological mean state of the storm track. In the following, we turn to this question by estimating the contribution of cyclones to the observed climatological differences between the experiments.

### 3.4 Contribution of cyclones to the climatological differences introduced by smoothing the SST

#### 3.4.1 Surface Heat Fluxes

In the CNTL climatology, we observe a maximum of latent and sensible surface heat fluxes on the warm side of both the Gulf Stream and the Kuroshio SST front (Fig. 7a,e). Peak latent heat fluxes are slightly offset to the south of the peak sensible heat

fluxes. TSSb associated this offset with the increase of sea surface temperature saturation mixing ratio with increasing SSTs following the Clausius-Clapeyron relation. In the North Atlantic, latent and sensible heat fluxes exceed 350 W m$^{-2}$ and 80 W m$^{-2}$, respectively (Fig. 7a). Sensible heat fluxes are slightly larger in the North Pacific, whereas latent heat fluxes remain slightly weaker (Fig. 7e).

The surface heat fluxes are similarly distributed in the ERA-Interim dataset, though latent heat fluxes in CNTL are considerably larger compared to ERA-Interim (compare Fig. 7a with Fig. S1 c, and Fig. 7e with Fig. S1 f). This discrepancy most likely arises from the difference of the SST resolution between the AFES and ERA-Interim, with the latter having a lower resolution prior to 2002 (Masunaga et al., 2015). Comparing the CNTL fluxes with the ERA-Interim winter climatology after 2002, the differences are significantly reduced (not shown).

The SST smoothing affects the amount of surface heat fluxes in both regions, though to a different extent. In the Gulf Stream region, we observe considerably weaker surface heat fluxes (Fig. 7b) along the weaker SST gradient (Fig. 2c). The maximum decrease of the latent heat fluxes is of the order of 120 W m$^{-2}$, while the reduction of sensible heat fluxes exceeds 30 W m$^{-2}$ (in line with Small et al. (2014)). A slight increase of surface heat fluxes is observed mainly to the northeast and less in the southern parts of the Gulf Stream region (Fig. 7b), consistent with the increase of SSTs due to the smoothing (Fig. 1c). This dipole of positive and negative anomalies of the surface heat fluxes is less pronounced in the Kuroshio region (Fig. 7f), reaching only about half the amplitude compared to the Gulf Stream region. We attribute the reduced amplitude to the smaller impact of the SST smoothing on the SST distribution in the Kuroshio region (Fig. 1f).

To estimate the role of cyclones for the differences when smoothing the SST, we decompose the winter climatology considering the surface heat fluxes occurring within and outside of a radius of 750 km around the cyclones' center over their entire lifetime (Fig. 7c,d,g,h). This decomposition allows us to assess how much of the climatological differences are associated with cyclones.

The climatological differences between CNTL and SMTHG/SMTHK predominantly arise when we do not consider heat fluxes associated with cyclones (Fig. 7c,d,g,h). On the other hand, the contribution from cyclones to the climatological heat flux differences is smaller in amplitude and the distribution of the flux anomalies less closely resembles the climatological differences between the experiments (Fig. 7b,f). The largest differences in Fig. 7d arise close to the climatological SST front position and are clearly connected to the presence or absence of a sharp SST front. This can be explained by a cold air mass transitioning over a SST front experiencing less surface heating when the smoothing reduced the SSTs on the warm side of the SST front in SMTHG and SMTHK (see Fig. 1c,f; Zolina and Gulev, 2003 and Vanniere et al., 2017b). The differences are more pronounced in the Gulf Stream region (Fig. 7c,d) compared to Kuroshio (Fig. 7g,h) due to the larger total mean differences of surface heat fluxes after the SST smoothing in the Atlantic (Fig. 7b,f).

Our results indicate that the smoothing of the SST front has only a minor effect when we consider cyclones, which is confirmed by a cyclone-relative composite analysis, where surface heat fluxes are only moderately reduced by the smoothing of the SST front (not shown). Our findings are also in line with Rudeva and Gulev (2011), who indicated that cyclones in the North Atlantic sector do not contribute significantly to the climatological surface heat fluxes in the region. Vanniere et al. (2017a) and Marcheggiani and Ambaum (2020) argued that the cold sector of cyclones contributes significantly to the air-sea

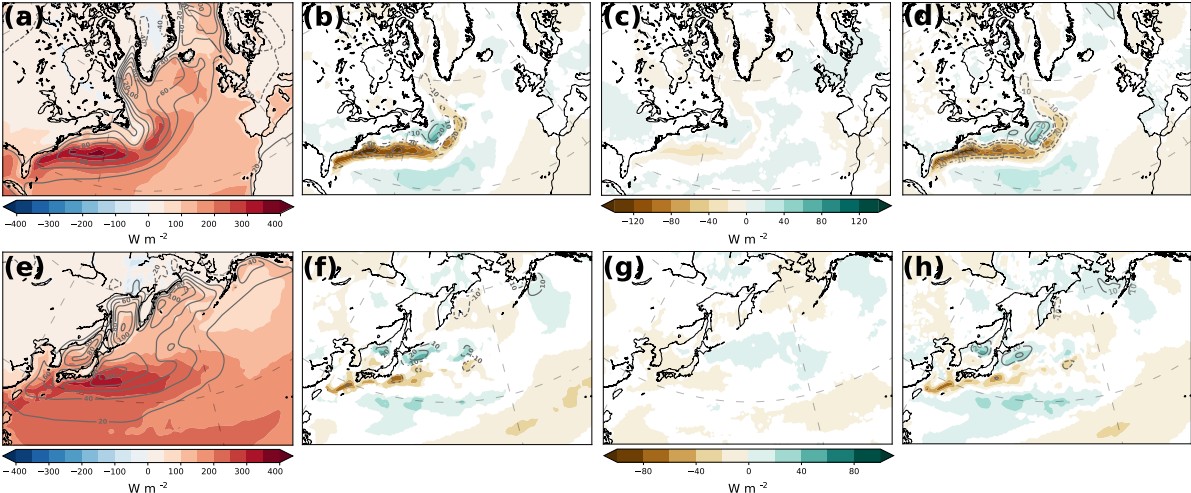

**Figure 7.** (a) Climatological latent (shading, W m$^{-2}$) and sensible heat fluxes (contours, W m$^{-2}$) for the North Atlantic for CNTL. (b) Statistically significant (>95%, based on the t-test) difference of the heat flux climatologies between SMTHG and CNTL (SMTHG-CNTL) for latent (shading) and sensible heat fluxes (grey contours, interval: 10 W m$^{-2}$, zero contour omitted). (c,d) As (b) but separated for fluxes within (c) and outside (d) a radius of 750 km around a cyclone centre. (e-h) As (a-d), but for the North Pacific.

heat exchange and thus to the re-establishment of low-level baroclinicity associated with the SST front. While they mainly associate the cold sector with synoptic activity, it can be argued that large parts of what they define as the cold sector reside outside of the cyclone area and cold air outbreaks featuring significant surface fluxes have been shown to also occur more distant to a cyclone core (Terpstra et al., 2021). Thus, we argue that their findings are consistent with ours and in line with Rudeva and Gulev (2011), with the fluxes in their cold sector being generally more distant to the central cyclone area.

### 3.4.2 Precipitation

The precipitation distribution in the Atlantic in CNTL is characterised by a maximum of large-scale ($> 6$ mm day$^{-1}$) and convective precipitation ($> 4$ mm day$^{-1}$) along the Gulf Stream SST front extending eastward (Fig. 8a). The maximum values of precipitation are located along the SST front (Fig. 3a), with a well-defined convective rain-band displaced towards the warm side of the Gulf Stream SST front, consistent with the findings of Kuwano-Yoshida et al. (2010b).

In the Pacific, we observe an analogous spatial distribution of the precipitation pattern, but the amplitude is somewhat larger than in the Gulf Stream region (compare Fig. 8a with Fig. 8e). Further, the peaks of large-scale and convective precipitation are more collocated in the western North Pacific compared to the North Atlantic. When compared to ERA-Interim (Fig. S2 a,d), there is a good resemblance of the spatial distribution of precipitation, but we notice higher large-scale and convective precipitation in AFES, consistent with the larger latent heat flux.

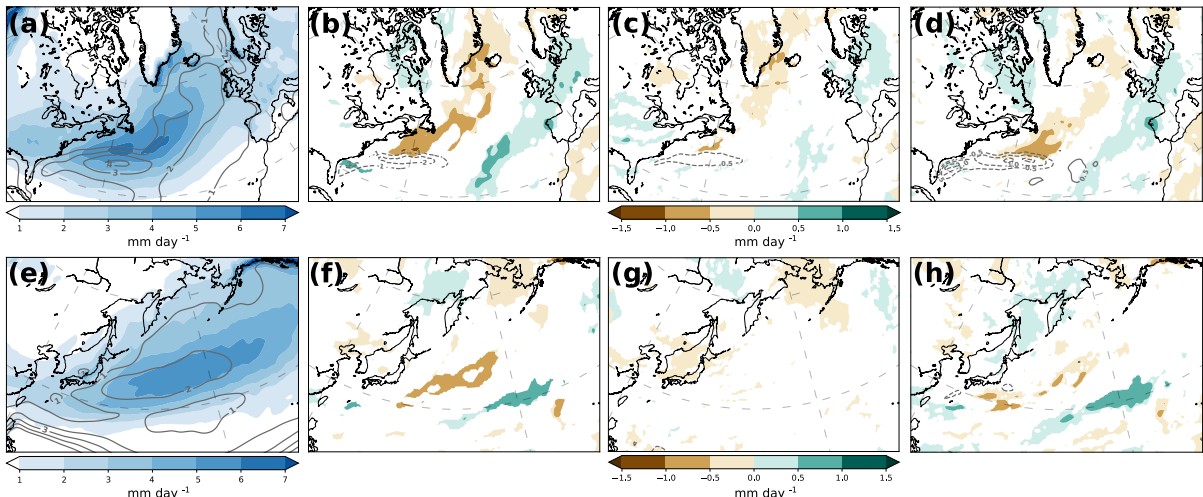

**Figure 8.** (a) Climatological large-scale precipitation (shading, mm day$^{-1}$) and convective precipitation (grey contours, interval: 1, mm day$^{-1}$, zero contour omitted) for the North Atlantic for CNTL. (b) Statistically significant (>95%) difference of precipitation climatologies between SMTHG and CNTL (SMTHG-CNTL) for large-scale (shading) and convective precipitation (grey contours, interval: 1, mm day$^{-1}$, zero contour omitted). (c,d) As (b) but separated for large-scale and convective precipitation within (c) and outside (d) a radius of 750 km around a cyclone centre. (e-h) As (a-d), but for the North Pacific.

Analogous to the surface heat fluxes, the smoothing of the SST field affects precipitation in both regions. In the North Atlantic, the smoothing leads to a remarkable decrease of precipitation (Fig. 8b), in line with the study of Kuwano-Yoshida et al. (2010b). In the North Pacific, however, precipitation is reduced considerably less, with both large-scale and convective
precipitation being reduced by less than 1 mm day$^{-1}$ (consistent with Zhang et al. (2020)). We relate the more pronounced reduction of precipitation in the Gulf Stream to the originally sharper SST gradient (Fig. 2). We also observe a similar dipole pattern with reduced precipitation over the Gulf stream and Kuroshio core as well as increased precipitation in the southeast of both regions (Fig. 8b,f). This equatorward shift in precipitation is consistent with the equatorward shift of the storm track in the smoothed experiments (Fig. 4).

Among the two types of precipitation, convective precipitation is more sensitive to the SST smoothing. In SMTHG, the mean convective precipitation is reduced by half compared to CNTL and the narrow convective precipitation band observed in CNTL largely disappears in SMTHG (not shown). This finding is in line with Minobe et al. (2008) and Kuwano-Yoshida et al. (2010b), who showed that the Gulf Stream SST front is crucial for the distribution and amount of convective precipitation and found convective precipitation to be significantly reduced after heavily smoothing the SST distribution in their simulations
with the same model.

Compared to the surface heat fluxes, the precipitation associated with cyclones is more influenced by the SST smoothing, in particular in SMTHG. There is a noticeable reduction in convective precipitation in the Gulf Stream region just south of the climatological position of the SST front (Fig. 8c), the region featuring the strongest decrease of SST due to the smoothing (Fig.

1c). This finding supports TSSa, who found convective precipitation to be closely related to the SSTs underneath the cyclone
core. Taking into account the precipitation, which is not associated with cyclones, the SST smoothing results in a reduction of
both large-scale and convective precipitation almost everywhere in the western North Atlantic (Fig. 8d).

Overall, cyclones account for a larger fraction of the precipitation differences than they did for the difference in surface
heat fluxes when comparing CNTL and SMTHG. This result is also in line with Hawcroft et al. (2012), who found the winter
precipitation in the Northern Hemisphere to be associated with extratropical cyclones. They also showed that the contribution
of cyclones in the Gulf Stream region accounts for 55-80% of the total DJF precipitation. This considerable fraction suggests
that precipitation should significantly shift equatorward along with the cyclone track density, which is consistent with our
analysis (Figs. 4a, 8c).

In the Kuroshio region we note a rather equal influence of the SST smoothing when we consider precipitation related or
unrelated with cyclones. These differences mainly concern the large-scale precipitation and are more evident in the central
North Pacific (Fig. 8h), forming a dipole of reduced precipitation to the north and increased precipitation to the south, similar
to the Atlantic (Fig. 8d). Interestingly, we note a slightly higher decrease of large-scale precipitation (approx. 0.5 mm day$^{-1}$)
when precipitation is not associated with cyclones in the vicinity of the Kuroshio, becoming less pronounced though when
increasing the radius from 750 km to 1000 km and thus including precipitation from features that are more distant from the
cyclone center (Fig. S8 d,f).

### 3.4.3 Specific Humidity at 850 hPa

Higher values of specific humidity are observed to the south of the Gulf Stream and Kuroshio regions (Fig. 9a,e) due to the
Clausius-Clapeyron relation with higher SSTs (Fig. 1a,d). The specific humidity maximum exceeds 6 g kg$^{-1}$ in the Gulf
Stream region, while lower maximum values (5 g kg$^{-1}$) are found in the southeastern part of the Kuroshio region (Fig. 9a,e).
The values are comparable to ERA-Interim (Fig. S2 b,e).
Analogous to the surface heat fluxes (Fig. 7b,f), smoothing the SST causes a noticeable decrease in specific humidity to the
south of the SST front (Fig. 9b,f), where the SSTs are lower than in CNTL (Fig. 1c,f). The decrease in specific humidity is
more pronounced in the Gulf Stream region, following the larger SST decrease in the Atlantic when the SSTs are smoothed.
Our findings are consistent with the results of Zhang et al. (2020), who analysed the meridional eddy specific humidity flux
instead of specific humidity.
For specific humidity, cyclones account only for a small part of the climatological differences between CNTL and SMTHG
(Fig. 9c). In addition to the larger amplitudes of the differences when specific humidity is not associated with cyclones (Fig.
9d), the structure is slightly different, with the negative anomalies being more zonally oriented than when considering the
contribution from cyclones (Fig. 9c). In the latter case, specific humidity is generally reduced in the Gulf Stream region,
whereas a slight increase is observed in the central part of the North Atlantic, most likely related to the southeastward shift of
the storm track after the SST smoothing (Fig. 9c and Fig. 4a). Regarding the specific humidity which is not related to cyclones,
we in contrast observe a slight increase to the north of the SST front, where the smoothing leads to an SST increase (Fig. 9d).
Consistently, the largest decrease evident in Fig. 9d towards the warm flank of the climatological SST front is clearly related to

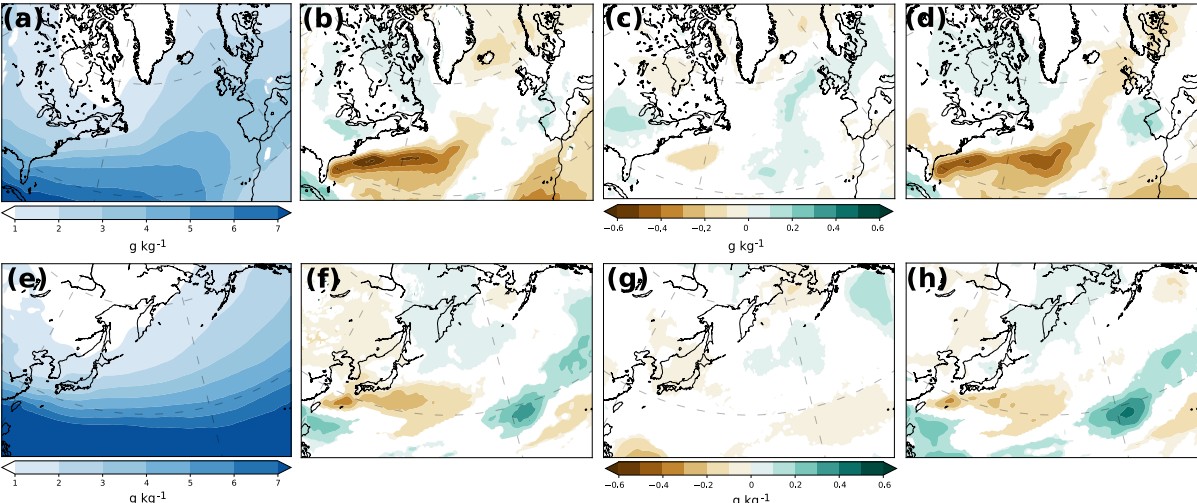

**Figure 9.** (a) Climatological specific humidity at 850 hPa (shading, g kg$^{-1}$) for the North Atlantic for CNTL. (b) Statistically significant (>95%) difference of climatological specific humidity between SMTHG and CNTL (SMTHG-CNTL). (c,d) As (b) but separated for specific humidity within (c) and outside (d) a radius of 750 km around a cyclone centre. (e-h) As (a-d), but for the North Pacific.

the largest decrease in the SST between SMTHG and CNTL (Fig. 1c), and the more pronounced decrease of specific humidity resembles the decrease in surface heat fluxes (Fig. 7d), when neglecting the contribution of cyclones for both variables.

Consistent with the results for the Atlantic, the North Pacific features larger differences in specific humidity, when the latter is not associated with cyclones propagating in the region (Fig. 9h). The location of a maximum decrease of approximately 0.3 g kg$^{-1}$ coincides with the region of the largest SST decrease (2K, Fig. 1f), located to the south of Japan. Apart from this reduction in specific humidity in the western part of the basin, there is an equivalent increase in specific humidity to the east, over the central North Pacific ocean (Fig. 9h), most likely triggering the increase of the large-scale precipitation observed in
this region (Fig. 8h).

     Apart from the well-established Clausius-Clapeyron relationship between SSTs and moisture, several studies indicate the leading role of cyclones on the poleward transport of moisture (e.g., Peixoto and Oort, 1992; Nakamura et al., 2004; Chang and Song, 2006; Newman et al., 2012). However, our results support instead that the SST is the dominating factor determining the distribution of specific humidity, with cyclones playing only a modulating role.

**3.4.4   Upper-level Wind Speed at 300 hPa**

In CNTL, the strongest climatological winds reach 40 m s$^{-1}$ in the Gulf Stream region and occur in a southwest (SW) to northeast (NE) tilted band (Fig. 10a) that is located close to the position of the SST front (Fig. 3a). In the Kuroshio region, the climatological jet is more zonal than in the North Atlantic, exceeds 60 m s$^{-1}$ (Fig. 10e), and is located somewhat to the south of the SST front (Fig. 3b). As for almost all the other fields, there is a good agreement between ERA-Interim and the AFES

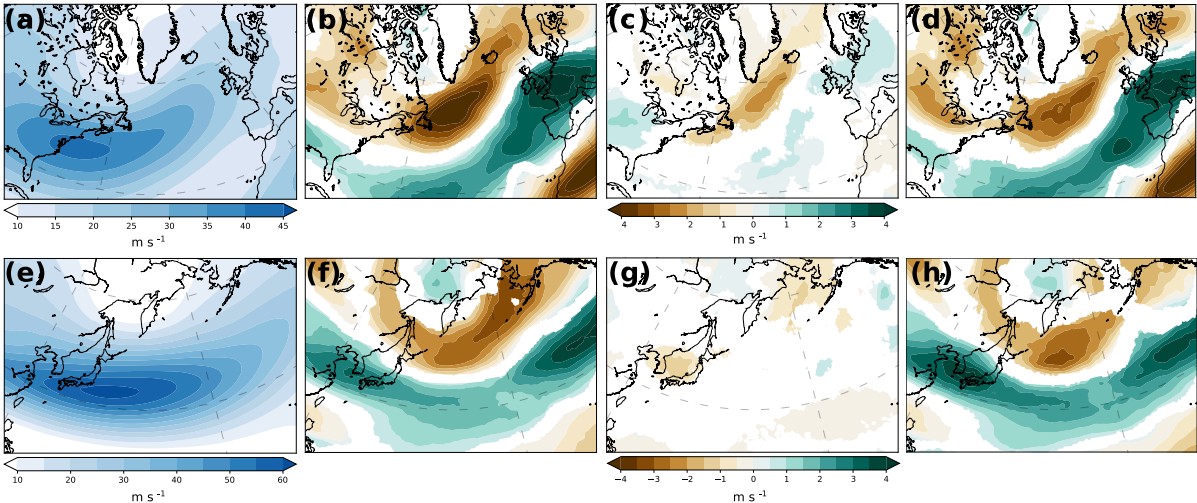

**Figure 10.** (a) Climatological wind speed at 300 hPa (shading, m s$^{-1}$) for the North Atlantic for CNTL. (b) Statistically significant (>95%) difference of the wind speed climatology between SMTHG and CNTL (SMTHG-CNTL). (c,d) As (b) but separated for wind speed within (c) and outside (d) a radius of 750 km around a cyclone centre. (e-h) As (a-d), but for the North Pacific.

simulations, with a slightly reduced maximum wind speed in the Gulf Stream region (approx. 5 m s$^{-1}$) and the maximum wind speed being geographically less extended over the North Pacific in the AFES climatology compared to ERA-Interim (compare Fig. 10a,e and Fig. S2 c,f).

    In both basins, we observe decreasing (increasing) wind speeds to the north (south) of the climatological jet position with smoother SSTs (Fig. 10b,f). This dipole is more pronounced downstream in the central and eastern North Atlantic and Pacific,
respectively. Kuwano-Yoshida and Minobe (2017) documented a similar difference pattern for the North Pacific jet.

    In the North Atlantic, the displacement of the maximum wind speed at 300 hPa is overall present for both when cyclones are present and absent. However, Fig. 10d explains more of the difference in the wind speed climatology (Fig. 10b), when compared to the wind speed associated with cyclones in the region (Fig. 10c). The different intensity of the displacement pattern partially reflects the fact that the region features slightly more time steps with no cyclones present (Fig. S13). In the North Pacific,
cyclones contribute even less (Fig. 10g) to the climatological differences between SMTHK and CNTL (Fig. 10f) compared to the respective results for the North Atlantic (Fig. 10c), with some differences though between the cyclone/no-cyclone patterns in the westernmost part of the Pacific (Fig. 10g,h).

    The higher contribution of cyclones to the observed differences in upper-level climatological wind speed in the Atlantic compared to the Pacific is consistent with previous studies indicating that the Pacific jet is externally (more thermally) driven,
as opposed to the Atlantic jet, which is more eddy-driven (e.g., Lee and Kim, 2003; Li and Wettstein, 2012).

## 4 Conclusions

We quantified and attributed differences in the atmospheric response when using realistic (CNTL) and smooth SSTs for the North Atlantic (SMTHG) and North Pacific (SMTHK), respectively, based on simulations with the AFES 3 model. The CNTL simulation compares well to ERA-Interim, except for considerably larger latent heat fluxes in CNTL, but these are most likely associated with the lower SST resolution in ERA-Interim prior to 2002. Overall, the AFES model successfully captured the distribution of pertinent variables in both ocean basins. Given the stronger SST gradient in the Atlantic, the effect of the smoothing on the SST front yields stronger SST differences between the CNTL and the respective smooth experiments (see Fig. 1c,f) as well as a distinctly stronger reduction of the SST gradient in the Atlantic compared to the Pacific (see Fig. 2c,f).

We first examined the impact of the smoothing of the SST on the intensification of individual cyclones. Considering only cyclones with a maximum intensification in the Gulf Stream or the Kuroshio region, we classified them into 3 categories based on their propagation relative to the SST front, where cyclones either always stay on the cold (C1) or warm (C2) side of the SST front or they cross the SST front from the warmer to the colder side (C3). Similar deepening rates for all these cyclone-categories across all experiments reveal the rather minor role of the SST gradient on the intensification of individual cyclones. This result is valid for both ocean basins, though it is particularly relevant for the Gulf Stream region where the SST smoothing dramatically weakens the strong SST gradient.

Considering all cyclones propagating in either the North Atlantic or the North Pacific, irrespective of their direction of propagation, stage of evolution, and their location of maximum intensification, we found the cyclone density in the storm track to decrease when the SSTs are smoothed in the Kuroshio and even more so in the Gulf Stream region. We relate the different response of the cyclone densities for the two regions to the more pronounced reduction of the SST gradient in SMTHG for the Atlantic compared to SMTHK for the Pacific. Overall, we observe an equator-ward shift in cyclone density in both regions, which is more pronounced over the central and eastern parts of the two ocean basins. Both cyclone density differences have a distinct SW-NE tilt, basically following the storm track. An analogous southward shift is observed in the upper-level jet for the North Atlantic, whereas for the North Pacific such a shift is absent and the difference between the experiments instead reveals a more meridionally focused and zonally extended jet with smoother SSTs.

We found a considerable decrease of both latent and sensible heat fluxes along the SST front when smoothing the SSTs, which was more pronounced across the SST front in the Gulf Stream region compared to the Kuroshio. Analogous to the surface heat fluxes, precipitation in the Gulf Stream region is strongly reduced when smoothing the SST front, which is particularly evident for convective precipitation on the warm side of the Gulf Stream SST front. However, both types of precipitation are only slightly affected by the SST smoothing in the Kuroshio region. Differences in specific humidity at 850 hPa feature a similar reduction after smoothing the SST. The weaker reduction of moisture and precipitation in the Kuroshio region is related to the smaller differences in the SST and SST front between CNTL and the smoothed fields in the Pacific compared to the Atlantic.

To clarify whether the differences between the CNTL and SMTH experiments stem directly from cyclones interacting with the SST and SST gradient, we considered selected variables within and outside an area with a radius of 750 km around cyclone

centres propagating in either the North Atlantic or the North Pacific. We found that the surface heat fluxes that are associated with cyclones in both basins do not considerably contribute to the climatological differences between the CNTL and SMTH experiments. Differences in precipitation, however, were more closely associated with cyclones propagating in either the North Atlantic or the North Pacific.

For specific humidity, cyclones have only a minor contribution to the climatological differences between CNTL and SMTHG/
SMTHK, with a more evident decrease in specific humidity in the Atlantic, arising from a considerable decrease in the SST gradient in the vicinity of the Gulf Stream. In contrast, both humidity and SST are not changed as significantly in the Pacific sector. Our results support that the underlying SST is the dominant factor determining the distribution of specific humidity, with cyclones playing a modulating role.

Similar to the surface heat fluxes and the specific humidity at 850 hPa, we found cyclones to only play a secondary role in
explaining the upper-level (300 hPa) wind speed differences arising from the SST smoothing. Notwithstanding this secondary role, Atlantic cyclones contribute more to the climatological differences than Pacific cyclones, which is consistent with previous studies indicating that the Atlantic jet is more eddy-driven than the Pacific jet.

Overall, our analysis highlights that SST fronts only have a minor impact on the characteristics and intensification of individual cyclones propagating in the Gulf Stream or Kuroshio region. Following the nomenclature of Haualand and Spengler
(2020), this indicates that the *direct* impact of the SST front on cyclone development is rather small, where *direct* is defined as the timely influence of surface heat fluxes on cyclone development occurring locally confined to the cyclone area. However, the SST front can have an *indirect* effect on the development of cyclones, where the altered SSTs reshape the large-scale environmental conditions in which cyclones form. Accordingly, we demonstrated that the differences in surface heat fluxes, specific humidity, convective precipitation, and upper-level wind speed between the CNTL and SMTH experiments largely arise in the
absence of cyclones. This *indirect* effect on cyclones is also consistent with recent findings on how atmospheric fronts interact with SST fronts (Reeder et al., 2021), where the *direct* impact of the SST front on atmospheric fronts by the SST front was found to be rather small. Thus, consistent with Reeder et al. (2021), the SST front mainly imprints itself onto the atmospheric climatology when no synoptic features are present.

*Code availability.* The codes to construct the figures in this study is available upon request.

*Data availability.* ERA-Interim data are provided by European Centre for Medium-Range Weather Forecasts (ECMWF) available online at https://www.ecmwf.int/en/forecasts/datasets/reanalysis-datasets/era-interim (Dee et al., 2011). The AFES data used in this study are publicly available at https://doi.org/10.11582/2021.00075.

*Author contributions.* LT carried out the bulk of the data analysis and writing. TS contributed to detailed discussion about the methods and interpretation of the findings as well as to the writing process. CS contributed to both data analysis and writing.

*Competing interests.* The authors declare that they have no conflict of interest.

*Acknowledgements.* We thank ECMWF for providing the ERA-Interim data and Akira Kuwano-Yoshida and Akira Yamazaki, who provided the AFES 3 data. We also want to thank Irina Rudeva and an anonymous referee for their constructive feedback. The AFES 3 data were supported by the Grant-in-Aid for Scientific Research on Innovative Areas 22106008 ("A 'hot spot' in the climate system: Extra-tropical air–sea interaction under the East Asian monsoon system") from the Ministry of Education, Culture, Sports, Science, and Technology of
Japan, and calculated using the Earth Simulator under JAMSTEC support. The work was conducted as part of the UNPACC project, funded by the Research Council of Norway (project number 262220).

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
