# Peer review of "Smoother versus sharper Gulf Stream and Kuroshio SST fronts: Effects on cyclones and climatology"

_Weather and Climate Dynamics, 2020_

## Referee Comment (RC1) · Irina Rudeva (Referee) · 10 Nov 2020

The paper explores the role that SST gradients in the Gulf Stream and Kuroshio play in extratropical cyclones in the two main storm tracks of the Northern Hemisphere. Surprisingly, the paper focuses mainly on the low-pressure systems that reach their maximum intensity over the Gulf Stream or Kuroshio, even though the largest changes in cyclone frequency in the experiments occur well downstream of either the Gulf Stream or Kuroshio (Fig.4).

My major comments are as follows:

[Figure]

1. The finding that cyclones play a secondary role in forming strong sensible and latent heat fluxes is only true for the low-pressure systems which have their centres in the selected areas, i.e. right over the oceanic currents. However, anomalous turbulent fluxes may not be associated with the cyclones with centres in the selected boxes. Tilinina et al. (2018; https://doi.org/10.1175/MWR-D-17-0291.1) showed that extreme fluxes over the Gulf Stream are linked to regions of cyclone-anticyclone interaction (usually associated with strong winds and large air temperature anomalies) when they are located above the ocean current. The finding by Tilinina et al. implies that the atmospheric circulation plays an important role in creating strong turbulent fluxes. While systems that have their centres within the selected boxes may, indeed, not be associated with strong air-sea fluxes, I believe that the paper should focus on a different subset of cyclones, i.e., those that are located $\sim$ 1000-3000 km to the north or north-east of the Gulf Stream or Kuroshio). This is in agreement with the area of the largest changes of cyclone activity shown in Fig.4.

2. I do not feel comfortable with the idea of using SST fronts from CNTRL for analysis of the smoothed runs. I believe that fronts should be objectively identified in all runs. This is particularly important in the Pacific, where SST fronts can hardly be seen in the the SMTHK experiments. As the classification of cyclones in this paper is based on the location relative to the SST front, in those cases when an SST front in SMTH cannot be detected, the type of cyclones cannot be defined. That said, given my first comment on the subset of cyclones that may be particularly affected by the the SST gradient, I am not sure that a classification based on the location of cyclone centres relative to the ocean current is particularly important. Fig. 5 shows that there are hardly any statistically significant differences between those types, not to mention the differences between SMTH and CNTRL runs.

3. Finally, it will be interesting to assess how much changes in heat fluxes, precipitation, etc. scale with a change in the SST gradient. It is mentioned multiple times that smaller reduction in the SST gradient in the Kuroshio region led to smaller corresponding changes in the atmospheric circulation. The question I want to raise here is if similar reductions in the SST gradient in those basins lead to comparable response. Or, alternatively, how stronger/weaker smoothing affects the same region.

Overall, the question raised in the paper ('how SST currents influence extratropical circulation?') is very intersting and existing simulations may shed light on it; however, in my opinion, the analysis should focus more on the systems downstream of the Gulf Stream and Kuroshio. Importantly, based on earlier publications and some plots in the submitted manuscript, I believe that such approach will change the main finding (and the title!) of the paper showing that SST gradients affect atmospheric cyclones.

Other comments:

Title: In the current version of the manuscript, it is, indeed, shown that the atmospheric conditions change stronger in the absence of cyclones over the ocean currents. However, as I said in the above, if more analysis is done on cyclones in the the storm track areas, the title may need to be changed.

l.90: Instead of just saying 'to the questions raised above', it is worth repeating the main questions here.

Data: I could not find the time resolution of the data used.

l.97: Why the period of integration ends in 2001? Considering the coarse SST resolution in ERA-Interim prior to 2002, selection of the time period needs to be justified.

l.114: Give a more detailed description of how the jet was detected.

l.104: Explain what is meant by "1-2-1 running mean"

l. 123: following a comment above, 3 consecutive time steps for the minimum life time of 12 hr comes from nowhere. I'd mention here that 12 hours is less than more often used threshold of 24 hours (i.e., five 6-hour time steps), applied in Neu et al. (2013; https://doi.org/10.1175/BAMS-D-11-00154.1) and adopted in many recent studies on

extratropical cyclones.

l.124, 144: I presume that you require that the cyclone centre, not just any point of the cyclone area, passes over the Gulf Stream or Kuroshio regions?

l.159: Also refer to Fig. 2a,b for the SST gradient.

Fig. 3: what are the units used to show the distribution of the SST fronts and jet axis? The caption says 'km of line/ 100km**2'. It suggests that the SST front is represented by a line, not by an area where the SST is above a 2 or 1.25K/100km (this indicates again, that the method used to define the SST front needs to be better described in Section 2.2, see my earlier comment). Explain how you got km of line per a unit area. Why not showing the frequency of the SST values above a threshold instead?

As I do not quite understand the units in Fig.3, I am not sure how to interpret higher jet distribution values over the Pacific. The text says that the jet is 'stronger', does that mean that it is present more often or that it is wavier?

l. 172: How do you know that it is the jet that affects the cyclones, not the other way round? More importantly, how does this section answers the question raised at the start of the paragraph (on the impact of the upper-level circulation on cyclones)? I only see a comparison of statistics at the upper and low levels.

l.179-187: As the Pacific jet is located further south compared to the Atlantic jet, it is hard to shift it more equatorward. Secondly, from my point of view, the magnitude of changes in two basins is very comparable despite of the smaller smoothing in the Pacific.

l. 188-191: Again, I do not see larger changes in the cyclone activity in the Atlantic. The patterns in the Atlantic and Pacific are slightly different, but this may be a result of the different location of the SST front/jet axis, as well as difference in the shape of continents. Moreover, I see a shifted storm track activity in the Atlantic or Pacific, however, basin-wide it does not seem to be weaker as opposed to the Southern ocean

mentioned in the text.

l.192: The differences in the Atlantic are also shifted between the upper and lower levels, less than in the Pacific, but in the same direction. It will be interesting to add cyclone frequencies in the middle of the troposphere (e.g., 500hPa).

Fig. 5: If you keep this plot in the revised manuscript, could you estimate if the difference between lines (i.e., between mean values for the different cyclone types at every time step) are statistically significant and mark the time steps when the mean value for a given type is significantly different from the other two types? I would recommend focusing the discussion on significant results.

l.250, Fig.6: I am not sure if Fig 6 is worth showing. As the cyclone classification in this paper is based on the location relative to the SST front, the mean SST in the cyclone area is somewhat prescribed.

l.256: In line with my earlier comments, the statement that weaker SST gradients in the Pacific are not strong enough to affect cyclone development, points to the fact that, perhaps, the SST threshold for the Pacific regions should be increased. Ideally, I think, they should be identical (even if the Atlantic threshold is decreased).

l.303: As I said in the major comments, cyclones that have centres over the oceanic currents are not expected to cause strong heat fluxes.

l. 307, Section 3.4.2: In line with my Major comment 1, it will be interesting to show how much precipitation changes with cyclones most affected by the SST change. (however, the increased precipitation over the ocean currents may be associated with another subset of cyclones, not those that are related to the increased heat fluxes)

l.313: Does smoothed SST in the Atlantic lead to a reduced separation distance between the maximum in large-scale and convective precipitation? If that does not happen, then explain why smaller SST gradients in the Pacific are responsible for more collocated precipitation maxima in the Pacific.

l.323: the largest shift in cyclones occurs downstream of the selected regions, however, the precipitation changes most significantly within the black boxes. I think it should be discussed in the text.

l.349: specific humidity can be affected by cyclones over the central part of the Notrh Atlantic. Besides local change in evaporation over the ocean currents, changes in characteristics of the extratropical cyclones downstream of the Gulf Stream and Kuroshio may modify the intensity of warm conveyor belts. There is also an increase in the relative humidity in the south-east corner of the Pacific basin, most likely associated with a southward shift of the storm track.

Fig. 7-10: Indicated statistical significance of the differences.

l.395: The pattern in the Atlantic (jet, cyclone frequency) suggests negative NAO in the smoothed runs.

l.400: I like the hypothesis here. However, in my opinion, the jet response in the Pacific is mainly associated with a strong increase in cyclone frequency in the Bay of Alaska (Fig.4c). A well developed cyclonic circulation will increase (decrease) the wind speed to the south(north) of its centre. This low pressure system may also be responsible for the moisture advection from the central Pacific to north-east (Fig. 9f). Interestingly, cyclones over the Kuroshio are associated with an intensified subtropical jet.

Minor comments:

l.6: I'd replace the 'intensification' with 'intensity', unless you really want to stress the process of deepening, rather than the maximum strength of the systems.

l.14: change 'states' to 'state'

p.1, l17: and throughout the paper: change to 'regions' as the Gulf Stream and Kuroshio are two independent regions (as opposed to, e.g., the Kara and Barents Sea)

l.102: Introducing SMTHG and SMTHK, it is worthwhile mentioning the Gulf Stream

and Kuroshio, respectively, instead of the North Atlantic and Pacific. Otherwise, it is not immediately obvious what K and G stand for.

l.145: replace 'intensification' with 'intensity'.

l.199: Perhaps, 'classification' sounds better.

l.341: Did you mean that precipitation should shit equatorward in case of a reduced SST gradient? l376: change to '40%'

It will be good to show lon/lat values for the gridlines (at least once in the beginning).

Fig. 1, caption: Remove the first '(a)' from the caption. In c (and the following figures), make clear that it is SMTHG minus CNTL. The caption should mention black solid lines.

Fig.2, b, e: Why did you choose to show the SST gradient in 0.5K/100km instead of K/100km?

Fig.7-10: I do not think I understand the last sentence in the captions. What kind of scaling was applied? In this case please describe in the Methods.

Fig. 3: On my screen the 'orange' colour scale looks nowhere near to orange, I'd call it pale red. Perhaps, change the scale a little bit.

Fig. A1, panels b and e: The range of values in the colour scale is too big and yellow shading is hard to see against the white background. The colour scale does not match either Fig. 2a or 2d, so I do not see why you chose such large range.
* * *

---

## Referee Comment (RC2) · Anonymous Referee #2 · 11 Nov 2020

This manuscript investigates the effect that SST smoothing of the Gulf Stream and Kuroshio has on extra-tropical cyclones reaching maximum intensity in the region, and attempts to explain whether the well-known impact of the SST smoothing on the mean state can be better understood when cyclones are present or not.

The topic is important and very worthwhile exploring, however I have some serious concerns regarding the scientific approach and interpretation of results. This includes the lack of consideration of some previous work directly relevant to this problem, some questions over the data and methods used, and a lack of testing regarding the sensitivity of the results to specific definitions. Furthermore, I unfortunately find many of the

conclusions to be unfounded. Specific comments are listed below, I put an (M) next to ones I consider major.

———

(M): The introduction provides a good comprehensive overview of many aspects of GS/KE -atmosphere interaction, but fails to discuss a large body of work directly relevant to the main focus of this paper i.e. SST influence on time-mean as affected by changes in synoptic storms. The following studies have already considered contributions of storms to the mean atmospheric state in the Gulf Stream and Kuroshio regions (Parfitt and Czaja, O'Neill et al, Parfitt and Seo, Masunaga et al, a,b). The first two of these studies show that the mean state in the tropospheric wind fields and precipitation are set by extreme synoptic situations, with the latter four studies specifically highlighting that it is the atmospheric fronts (and not really the cyclones). Indeed, when you remove the atmospheric fronts from the climatology, you remove the time-mean convergence signal above the Gulf Stream and Kuroshio altogether (see Figure 2 in Parfitt and Seo) as the remaining baroclinic components mostly cancel out (see Figure 3 and Supplementary Figure 2 in Parfitt and Seo). Furthermore, atmospheric fronts contribute ∼90% of the rainfall in these regions (Catto et al., 2016).

One of your main conclusions (that changes in individual cyclones resulting from your SST changes aren't important for changes in the mean state) is therefore not surprising given the results of these aforementioned studies (that show the mean state in both winds and precipitation is set by atmospheric fronts). I think the authors should discuss these studies, as well as the consistency of their results with them. Furthermore, I think it is fair (and scientifically interesting) to request that the authors make some attempt to use these studies together with their results to make some additional hypothesis regarding the relationship between SST front, cyclone and atmospheric front (e.g. SST frontal influence on mean-state mostly through weak atmospheric fronts/weak cyclones, as discussed in Masunaga et al, 2020b, or some other idea).

Parfitt, R., & Czaja, A. (2016). On the contribution of synoptic transients to the mean atmospheric state in the Gulf Stream region. Quarterly Journal of the Royal Meteorological Society, 142(696), 1554-1561. O'Neill, L. W., Haack, T., Chelton, D. B., & Skyllingstad, E. (2017). The Gulf Stream convergence zone in the time-mean winds. Journal of the Atmospheric Sciences, 74(7), 2383-2412. Parfitt, R., & Seo, H. (2018). A New Framework for Near‐Surface Wind Convergence Over the Kuroshio Extension and Gulf Stream in Wintertime: The Role of Atmospheric Fronts. Geophysical Research Letters, 45(18), 9909-9918 Masunaga, R., Nakamura, H., Taguchi, B., & Miyasaka, T. (2020a). Processes Shaping the Frontal-Scale Time-Mean Surface Wind Convergence Patterns around the Kuroshio Extension in Winter. Journal of Climate, 33(1), 3-25. Masunaga, R., Nakamura, H., Taguchi, B., & Miyasaka, T. (2020b). Processes shaping the frontal-scale time-mean surface wind convergence patterns around the Gulf Stream and Agulhas Return Current in winter. Journal of Climate, 33(21), 9083-9101. Catto, J. L., Jakob, C., Berry, G., & Nicholls, N. (2012). Relating global precipitation to atmospheric fronts. Geophysical Research Letters, 39(10).

Line 80: Following on from the above, the strong surface fluxes primarily occur behind cold fronts that bring cold dry air off the continent, not necessarily with cyclones as a whole (especially when the cyclones are positioned directly over the GS), and are modulated thus. Also, this paragraph should clearly reference Bishop et al. (2017) which discusses the scale dependence of SSTs and heat fluxes in the western boundary current regions in terms of atmosphere-driven vs. ocean-driven.

Bishop, S. P., Small, R. J., Bryan, F. O., & Tomas, R. A. (2017). Scale dependence of midlatitude air–sea interaction. Journal of Climate, 30(20), 8207-8221.

(M): I have a few serious concerns about the data used for this study. Firstly, it has been shown many times how important model resolution is for accurately resolving air-sea interaction in western boundary current regions. Some examples: Smirnov et al. (2015) show the necessity for both ocean and atmosphere at 0.25deg in order for SST-induced heating to be balanced by transients rather than cold-air advection. At

50km, you are not there, and barely resolving the cross-frontal scale in the atmosphere (∼100km). It is also well known that you need eddy-resolving (∼1/10deg) resolution in the ocean to actually resolve many of the oceanic eddy processes crucial to the GS/KE air-sea interaction (e.g. Figure 2 from Hewitt et al., 2017). These eddies are known to be crucial for the interaction (Ma et al., 2015) and its influence on general atmospheric and oceanic variability. Your resolution falls under these values and I don't see how definitive conclusions can be made given we already know the interaction cannot be fully resolved. The importance of resolution and resolving transients for the hemispheric flow is also shown in Lee et al. (2018). I also can't understand why ERA-Interim is used instead of ERA-5 which is available, which has a much better resolution atmosphere and ocean. You even note this yourself in Line 157. Please also mention in Section 2.1 what temporal resolution you use. How often do you calculate SST fronts etc?

Smirnov, D., Newman, M., Alexander, M. A., Kwon, Y. O., & Frankignoul, C. (2015). Investigating the local atmospheric response to a realistic shift in the Oyashio sea surface temperature front. Journal of Climate, 28(3), 1126-1147. Hewitt, H. T., Bell, M. J., Chassignet, E. P., Czaja, A., Ferreira, D., Griffies, S. M., ... & Roberts, M. J. (2017). Will high-resolution global ocean models benefit coupled predictions on short-range to climate timescales?. Ocean Modelling, 120, 120-136. Ma, X., Chang, P., Saravanan, R., Montuoro, R., Hsieh, J. S., Wu, D., ... & Jing, Z. (2015). Distant influence of Kuroshio eddies on North Pacific weather patterns?. Scientific reports, 5, 17785. Lee, R. W., Woollings, T. J., Hoskins, B. J., Williams, K. D., O'Reilly, C. H., & Masato, G. (2018). Impact of Gulf Stream SST biases on the global atmospheric circulation. Climate Dynamics, 51(9-10), 3369-3387.

Line 110 - What is the sensitivity of your results to choice of threshold on the SST gradient? Also, how can you confidently make comparisons between the Atlantic and the Pacific when you have a different threshold for both?

Line 120 - What about the sensitivity to cyclone detection? Neu et al. clear show

vastly different climatologies depending on the specific method. Neu, U., Akperov, M. G., Bellenbaum, N., Benestad, R., Blender, R., Caballero, R., ... & Grieger, J. (2013). IMILAST: A community effort to intercompare extratropical cyclone detection and tracking algorithms. Bulletin of the American Meteorological Society, 94(4), 529-547.

Line 123-125 - This seems like a strange and unnecessary choice to me, given the importance of quasi-stationary systems in the region (Masunaga et al., 2020)

(M): Section 2.4. I see this as a serious concern. Firstly, I have a problem with using the SST front definition from the CNTL experiment to define the location in the SMTH. The fact that there are no "fronts" in the SMTH points to a shortcoming in the methodology being used to identify them. Also, the definition of SST front in the CNTL experiment is also concerning, as it is simply based on an SST gradient. The SST front is not a straight line (I suggest you look at a figure like Figure 1 from Andres (2016)) - how do you deal with Gulf Stream rings for example? This also makes me rather skeptical of the definitions of C1, C2 and C3, despite their use in previous manuscripts. Additionally, if there is no defined SST front in the SMTH experiment, then there is minimal relevance to the definitions of C1, C2, and C3 anyway. Andres, M. (2016). On the recent destabilization of the Gulf Stream path downstream of Cape Hatteras. Geophysical Research Letters, 43(18), 9836-9842.

Figure 3: It is not clear what I am looking at here - also, how do you arrive at the units? Also, please provide further information on the jet detection method.

Line 185 - In my opinion, this hypothesis should be removed as it implicitly assumes an SST influence on the jet, which is meant to be a topic of exploration here. Similar comments apply to Line 188 and Line 288.

Figure 4: I find the wording of this caption extremely confusing.

(M): Line 200 - I cannot understand the logic behind not just looking at all cyclones. Line

223 - you make the bold claim that cyclone intensification is only weakly mofidied by SST gradient, but what about all those that aren't undergoing maximum intensification right there?

(M): Line 258 - This is a main point of the paper and one I don't feel comfortable with. Your claim is that the SST gradient is not particularly important for the intensification of individual cyclones. Whilst I may agree with this general sentiment (see my first major comment), I do not think the analysis presented here can make that conclusion. 1) As mentioned previously, you are looking at a subset only. 2) There are many differences between your two experiments, not just the SST gradient. The absolute SST changes also, and I strongly suspect that the variability in the Gulf Stream path length/separation from coast changes too, each having been shown to significantly affect the interaction. I don't see anything up to this point that allows you to definitively link it to the SST gradient. You mention in Line 40 that these factors need to be teased apart, but I am not convinced that has been done here simply by defining C1, C2 and C3 - these classifications are also ill-defined in the SMTH experiment. Line 266 - as I alluded to earlier, it does not seem appropriate to use two different subsets of cyclones for each of these sections (individual vs. mean state) if you are going to draw comparisons between them. Additionally, given that the typical timescale for thermal air-sea interaction is $\sim$ a day or so, one would expect the GS/KE to impact some storms further downstream from the region. This again raises questions regarding the conclusion that the SST front does not affect individual cyclones, just because the ones reaching maximum intensity in that specific regions are not that affected.

(M): Section 3.4. Line 294 -Parfitt and Czaja (2016) have shown that the top 30% of climatological latent heat fluxes in the Gulf Stream region are associated with cyclones that have just passed over the region. It is continuously said here "when no cyclones are present in the respective region", however that simply means the cyclone center is not directly above the region - this does not mean a cyclone's passing isn't responsible for the trailing cold air that maximises the fluxes. This is relevant for consistency with

Section 3.4.2 later, where it is shown cyclone precipitation does change significantly. A lot of moisture availability, that is ultimately taken up in say the warm conveyer belt of any given cyclone, in these regions can come from the strong latent heat fluxes behind the cold front of a system that passed ahead of it. This comment is also relevant for your statement "for specific humidity, cyclones account only for a small part of the climatological differences" in Line 359, and for your statement from Line 379-383. In particular, I do not agree with Line 381-383. This comment feeds back to the previous one, where I do not think conclusions about the SST influence on the atmosphere through cyclones can be drawn simply by comparing whether a cyclone is in that specific region or not.

Figure 8: The colors on Figure 8 (b-h) I would recommend changing as it is hard to see any difference.

Section 3.4.4 - I would recommend mentioning Lee et al. (2018) here also.

Lastly, I noticed a number of spelling/grammatical errors, I suggest a thorough check. (e.g. line 168, dipolar etc.).

---

## Author Comment (AC1) · 19 Dec 2020

**Response to Reviewer 1**

We sincerely thank reviewer Irina Rudeva for her constructive critique and detailed feedback of our manuscript. She addressed important and valid points that helped us to improve the manuscript. We provide detailed responses with our comments and changes in blue.

**MAJOR COMMENTS**

**1. The finding that cyclones play a secondary role in forming strong sensible and latent heat fluxes is only true for the low-pressure systems which have their centres in the selected areas, i.e. right over the oceanic currents. However, anomalous turbulent fluxes may not be associated with the cyclones with centres in the selected boxes. Tilinina et al. (2018; https://doi.org/10.1175/MWR-D-17-0291.1) showed that extreme fluxes over the Gulf Stream are linked to regions of cyclone-anticyclone interaction (usually associated with strong winds and large air temperature anomalies) when they are located above the ocean current. The finding by Tilinina et al. implies that the atmospheric circulation plays an important role in creating strong turbulent fluxes. While systems that have their centres within the selected boxes may, indeed, not be associated with strong air-sea fluxes, I believe that the paper should focus on a different subset of cyclones, i.e., those that are located ∼ 1000-3000 km to the north or north-east of the Gulf Stream or Kuroshio). This is in agreement with the area of the largest changes of cyclone activity shown in Fig.4.**

We agree with the reviewer that the attribution to only cyclones in the box might have been too restrictive and we will complement our previous analysis with one in which we will consider the pertinent variables occurring within different radii of any cyclone over their entire lifetime in the ocean basin irrespective of its location. A preliminary analysis indicates that this additional analysis will support our previous conclusions, but we will adapt the manuscript accordingly should they not.

**2. I do not feel comfortable with the idea of using SST fronts from CNTRL for analysis of the smoothed runs. I believe that fronts should be objectively identified in all runs. This is particularly important in the Pacific, where SST fronts can hardly be seen in the the SMTHK experiments. As the classification of cyclones in this paper is based on the location relative to the SST front, in those cases when an SST front in SMTH cannot be detected, the type of cyclones cannot be defined. That said, given my first comment on the subset of cyclones that may be particularly affected by the the SST gradient, I am not sure that a classification based on the location of cyclone centres relative to the ocean current is particularly important. Fig. 5 shows that there are hardly any statistically significant differences between those types, not to mention the differences between SMTH and CNTRL runs.**

We thank the reviewer for her comment, showing that our rationale for this choice has not become apparent in the current state of the manuscript. We use this classification only to be able to compare cyclones with a geographically similar genesis location and track across the experiments. We will adapt the manuscript to point this out more clearly. Having said this, we did attempt to objectively identify the SST fronts for the SMTH experiments, but this fails because the SST gradient is very homogeneous over a large region such that the detection of the SST front location is mostly determined by numerical noise rather than a defined maximum in the SST gradient.

**3. Finally, it will be interesting to assess how much changes in heat fluxes, precipitation, etc. scale with a change in the SST gradient. It is mentioned multiple times that smaller reduction in the SST gradient in the Kuroshio region led to smaller corresponding changes in the atmospheric circulation. The question I want to raise here is if similar reductions in the SST**

**gradient in those basins lead to comparable response. Or, alternatively, how stronger/weaker smoothing affects the same region.**

We thank the reviewer for raising this issue. As mentioned in the "Data and Methods" section, we use data from model experiments that were conducted at and provided by JAMSTEC. For this reason, we unfortunately do not have the capability to run additional experiments to estimate how stronger/weaker smoothing affects the same region (the Gulf Stream or the Kuroshio region). Nonetheless, we thank the reviewer for sharing with us this interesting idea, which is an interesting potential analysis in the future, when we will be capable to execute more/different experiments.

**OTHER/MINOR COMMENTS**

We thank the reviewer for all the "other/minor comments" she raised and we will consider them when editing the manuscript. Please let me answer/clarify some of these questions.

**Title: In the current version of the manuscript, it is, indeed, shown that the atmospheric conditions change stronger in the absence of cyclones over the ocean currents. However, as I said in the above, if more analysis is done on cyclones in the the storm track areas, the title may need to be changed.**

In this version of our manuscript, we strictly associate our results to the Gulf Stream and Kuroshio regions, as these are defined in the manuscript. As the reviewer underlined, for these regions "it is indeed shown that the atmospheric conditions change stronger in the absence of cyclones over the ocean currents". Based on a preliminary analysis, we believe that our additional analysis (see our response to Major comment 1) will confirm our previous findings and will allow us to extend this argument to the wider area of the North Atlantic and North Pacific, respectively. Of course, if needed we will edit/change the title to correspond to our new findings.

**l.97: Why the period of integration ends in 2001? Considering the coarse SST resolution in ERA-Interim prior to 2002, selection of the time period needs to be justified.**

As indicated previously (please see our response to your Major comment 3), the model experiments we use were conducted for the period 1982-2001 and this is why we are not able to extend this time step after 2002. But this is not a caveat, as the AFES simulations do not suffer from the same limitation as ERA-Interim, but use high-resolution SSTs as boundary conditions for the entire time period.

**l.104: Explain what is meant by "1-2-1 running mean"**

It is a three-point filter with the weights 0.25, 0.5 and 0.25. The 1-2-1 filter has a sharp cutoff, so that unwanted frequency components are effectively removed. This filter was applied to the data prior to JAMSTEC providing it to us. For further information please see Kuwano-Yoshida and Minobe, 2017 (https://journals.ametsoc.org/view/journals/clim/30/3/jcli-d-16-0331.1.xml?tab_body=fulltext-display).

**l. 123: following a comment above, 3 consecutive time steps for the minimum life time of 12 hr comes from nowhere. I'd mention here that 12 hours is less than more often used threshold of 24 hours (i.e., five 6-hour time steps), applied in Neu et al. (2013; https://doi.org/10.1175/BAMS-D-11-00154.1) and adopted in many recent studies on extratropical cyclones.**

We thank the reviewer for letting us clarify this issue. Following Neu et al. (2013) we indeed followed the same and often used technique, considering cyclone tracks with at least five 6-hour time steps, but in addition we required cyclones to have three 6-hour time steps in the Gulf Stream or the

Kuroshio region in order to classify the cyclones into different categories (C1-5). Asking for five 6-hour time steps only in the "box"/area of interest would significantly reduce the total number of tracks. Given the geographically restricted region and considering that the specific areas are regions of cyclogenesis, a further restriction would downgrade the significance of our findings. We will add this missing information in the manuscript.

**l.124, 144: I presume that you require that the cyclone centre, not just any point of the cyclone area, passes over the Gulf Stream or Kuroshio regions?**

Exactly. Please see our response on Major comment 1 to see how we will proceed with the new analysis.

**Fig. 3: what are the units used to show the distribution of the SST fronts and jet axis? The caption says 'km of line/ 100km**2'. It suggests that the SST front is represented by a line, not by an area where the SST is above a 2 or 1.25K/100km (this indicates again, that the method used to define the SST front needs to be better described in Section 2.2, see my earlier comment). Explain how you got km of line per a unit area. Why not showing the frequency of the SST values above a threshold instead? As I do not quite understand the units in Fig.3, I am not sure how to interpret higher jet distribution values over the Pacific. The text says that the jet is 'stronger', does that mean that it is present more often or that it is wavier?**

The reviewer is right. We do detect SST front lines rather than regions of strong SST gradients. We require front lines in order to be able to define when cyclones cross the front. We will point this out more clearly in the Methods section.

We acknowledge that the unit of length per area is somewhat unintuitive. We nevertheless use this unit when compiling a climatology, because this is independent of the grid resolution, while for example a naive masking of all grid points that contain a front line would yield lower detection frequencies the finer the analysis grid becomes.

**l.256: In line with my earlier comments, the statement that weaker SST gradients in the Pacific are not strong enough to affect cyclone development, points to the fact that, perhaps, the SST threshold for the Pacific regions should be increased. Ideally, I think, they should be identical (even if the Atlantic threshold is decreased).**

We thoroughly tested different thresholds to obtain reasonable SST fronts in both basins. Given the different SST gradient in the two basins (e.g., Nakamura et al., 2004; Tsopouridis et al., 2020b), using the Atlantic threshold, almost no SST fronts would have been captured in the Kuroshio region. Vice versus, with the Pacific threshold in the Gulf Stream region we would have obtained many weak SST fronts that would have made it difficult to identify the main SST front. We thus believe that a different threshold is necessary to accommodate the different natures of the boundary currents and SST fronts. We will adapt the text accordingly, to clarify the use of different thresholds in the two ocean basins.

**Fig. 7-10: Indicated statistical significance of the differences.**

We thank the reviewer for raising this issue. In the revised version of the manuscript, we will provide the statistically significant results, as the reviewer suggested.

---

## Author Comment (AC2) · 19 Dec 2020

**Response to Reviewer 2**

We sincerely thank the anonymous reviewer for his/her feedback and comments, which helped us to improve the manuscript. We provide detailed responses with our comments and changes in blue.

**(M): The introduction provides a good comprehensive overview of many aspects of GS/KE - atmosphere interaction, but fails to discuss a large body of work directly relevant to the main focus of this paper i.e. SST influence on time-mean as affected by changes in synoptic storms. The following studies have already considered contributions of storms to the mean atmospheric state in the Gulf Stream and Kuroshio regions (Parfitt and Czaja, O'Neill et al, Parfitt and Seo, Masunaga et al, a,b). The first two of these studies show that the mean state in the tropospheric wind fields and precipitation are set by extreme synoptic situations, with the latter four studies specifically highlighting that it is the atmospheric fronts (and not really the cyclones). Indeed, when you remove the atmospheric fronts from the climatology, you remove the time-mean convergence signal above the Gulf Stream and Kuroshio altogether (see Figure 2 in Parfitt and Seo) as the remaining baroclinic components mostly cancel out (see Figure 3 and Supplementary Figure 2 in Parfitt and Seo). Furthermore, atmospheric fronts contribute ~90% of the rainfall in these regions (Catto et al., 2016).**
**One of your main conclusions (that changes in individual cyclones resulting from your SST changes aren't important for changes in the mean state) is therefore not surprising given the results of these aforementioned studies (that show the mean state in both winds and precipitation is set by atmospheric fronts). I think the authors should discuss these studies, as well as the consistency of their results with them. Furthermore, I think it is fair (and scientifically interesting) to request that the authors make some attempt to use these studies together with their results to make some additional hypothesis regarding the relationship between SST front, cyclone and atmospheric front (e.g. SST frontal influence on mean-state mostly through weak atmospheric fronts/weak cyclones, as discussed in Masunaga et al, 2020b, or some other idea).**

We thank the reviewer for this comment and we will include these papers in the introduction and discussion sections of the revised manuscript. In particular the findings of Masunaga et al., 2020a,b are very much in line with our findings as well as the findings in a related submitted study focusing on the interaction of atmospheric fronts with SST gradients (Reeder et al., resubmitted to JAS). We would, however, caution whether what Masunaga et al., 2020a,b refer to as "weak systems" is representative of cyclones. We will thus complement the discussion presented in Masunaga et al. 2020a,b with our cyclone-specific viewpoint.
As fronts are specifically discussed in a related submission, we will not duplicate this analysis and discussion here. Following concerns from both reviewers, we will however complement our previous analysis with one in which we will consider climatologies of the pertinent variables within different radii of all cyclones in our database, irrespective of the location of the centre and their state in the life cycle. We vary radii between 500 and 1000 km, consistent with a maximum cyclone circumference in Wernli and Schwierz (2006) and the analysis of Rudeva and Gulev (2011). With the larger radii, we will thus also implicitly include the contribution of nearly all fronts in the analysis. Preliminary analysis supports our previous conclusions.

**Line 80: Following on from the above, the strong surface fluxes primarily occur behind cold fronts that bring cold dry air off the continent, not necessarily with cyclones as a whole (especially when the cyclones are positioned directly over the GS), and are modulated thus. Also, this paragraph should clearly reference Bishop et al. (2017) which discusses the scale dependence of SSTs and heat fluxes in the western boundary current regions in terms of atmosphere-driven vs. ocean-driven.**

**Bishop, S. P., Small, R. J., Bryan, F. O., & Tomas, R. A. (2017). Scale dependence of midlatitude air–sea interaction. Journal of Climate, 30(20), 8207-8221**

Following our response to the previous comment, we believe that a large enough radius would include the cold sector associated with the parent cyclone. However, cold air outbreaks must not necessarily occur in the cold sector of a cyclone, but can also be associated with an anticyclone (e.g., Dalavalle and Bosart 1975; Colucci and Davenport 1987; Nakamura et al., 2016). Hence, climatological features associated with these type of cold air outbreaks should not be accounted for in a cyclone specific climatology.

We thank the reviewer for pointing us to the paper by Bishop et al. (2017). We acknowledge this debate on whether the SST variability in the WBCs is atmospheric or ocean-driven on different timescales. However, with our analysis based on simulations with prescribed SSTs we will not be able to contribute to this debate and would thus find it misleading to discuss our results in the context of Bishop et al. (2017) in our manuscript.

**(M): I have a few serious concerns about the data used for this study. Firstly, it has been shown many times how important model resolution is for accurately resolving air-sea interaction in western boundary current regions. Some examples: Smirnov et al. (2015) show the necessity for both ocean and atmosphere at 0.25deg in order for SST-induced heating to be balanced by transients rather than cold-air advection. At 50km, you are not there, and barely resolving the cross-frontal scale in the atmosphere (~100km). It is also well known that you need eddy-resolving (~1/10deg) resolution in the ocean to actually resolve many of the oceanic eddy processes crucial to the GS/KE air-sea interaction (e.g. Figure 2 from Hewitt et al., 2017). These eddies are known to be crucial for the interaction (Ma et al., 2015) and its influence on general atmospheric and oceanic variability. Your resolution falls under these values and I don't see how definitive conclusions can be made given we already know the interaction cannot be fully resolved. The importance of resolution and resolving transients for the hemispheric flow is also shown in Lee et al. (2018). I also can't understand why ERA-Interim is used instead of ERA-5 which is available, which has a much better resolution atmosphere and ocean. You even note this yourself in Line 157. Please also mention in Section 2.1 what temporal resolution you use. How often do you calculate SST fronts etc?**

Regarding the first point of the reviewer on the model resolution, we would like to make three remarks. First, the AFES simulations on which we based our analysis have been used for numerous studies on the effect of SST gradients and fronts on the atmosphere (e.g., Nakamura et al., 2008; Kuwano-Yoshida et al., 2010; Kuwano Yoshida and Minobe 2017). We thus regard this as valid data in the context of previous studies on this topic.

Second, we appreciate and are aware of that smaller-scale ocean features can affect the climatological air-sea exchange to some extent. We here, however, aim to contribute to a discussion of a much more fundamental question, namely which processes are typically associated with the atmosphere air-sea exchange. It seems implausible that ocean eddies will fundamentally change these attributions and thus our conclusions.

Finally, our analysis based on this type of coarser model data is highly relevant to the climate modelling community, which typically uses even coarser resolutions. We will adapt the discussion and conclusions to make this focus of our analysis clearer.

Regarding ERA-Interim, we included this comparison to the AFES climatology only to make plausible that the AFES simulations, which are comparable in resolution to ERA-Interim, provide a reasonable depiction of the air-sea exchange and storm track over our target region. We have used ERA-Interim at 0.5° resolution, interpolated from the original 0.7°. In our department, we would only have access to ERA5 data at 0.5° resolution data due to constraints on data storage. Moreover, we have tested the sensitivity of our results to either just using 1979-2001 or 2002-2016, with no

significant changes between the two periods despite the resolution change in SST for ERA-Interim. Overall, ERA-Interim has been extensively and fruitfully used in numerous climatological analyses of air-sea interactions, so it remains a relevant data set.

Finally, regarding the SST fronts, we perform the detection using SST data filtered with a spectral truncation to T84 resolution and detect the SST fronts in the instantaneous SST field every 6-hours. We will add the missing information in the revised version of the manuscript.

**Line 110 - What is the sensitivity of your results to choice of threshold on the SST gradient? Also, how can you confidently make comparisons between the Atlantic and the Pacific when you have a different threshold for both?**

We thoroughly tested different thresholds to obtain reasonable SST fronts in both basins. Given the different SST gradient in the two basins (e.g., Nakamura et al., 2004; Tsopouridis et al., 2020b), using the Atlantic threshold, almost no SST fronts would have been captured in the Kuroshio region. Vice versus, with the Pacific threshold in the Gulf Stream region we would have obtained many weak SST fronts that would have made it difficult to identify the main SST front. We thus believe that a different threshold is necessary to accommodate the different natures of the boundary currents and SST fronts. We will adapt the text accordingly, to clarify the use of different thresholds in the two ocean basins.

**Line 120 - What about the sensitivity to cyclone detection? Neu et al. clear show vastly different climatologies depending on the specific method. Neu, U., Akperov, M. G., Bellenbaum, N., Benestad, R., Blender, R., Caballero, R., ... & Grieger, J. (2013). IMILAST: A community effort to intercompare extratropical cyclone detection and tracking algorithms. Bulletin of the American Meteorological Society, 94(4), 529-547**

We conducted several sensitivity experiments where we varied parameters in the detection and tracking namelists used by the algorithm. As the reviewer mentioned, there are qualitative differences between previous studies, but unfortunately no sufficient information is provided regarding the parameters used in each study. Motivated by this gap, we decided to publish our detection and tracking namelist in Tsopouridis et al., 2020a. The cyclone density pattern is in good spatial agreement with previous studies (e.g., Hanley and Caballero, 2012; Neu et al., 2013), successfully capturing the major regions of cyclone activity in both basins. We observe small quantitative differences compared to the density climatology presented by both Neu et al. (2013) or Murray and Simmonds (1991a), who also used the Melbourne University algorithm. These small deviations are most likely due to the neglect of shallow and weak systems in our database.

**Line 123-125 - This seems like a strange and unnecessary choice to me, given the importance of quasi-stationary systems in the region (Masunaga et al., 2020)**

We only consider actively developing cyclones in the region and further investigate their evolution and the associated mechanisms. Moreover, the steep orography around Greenland's coast would result in artefacts in the detection and therefore we do not include quasi-stationary systems.

**(M): Section 2.4. I see this as a serious concern. Firstly, I have a problem with using the SST front definition from the CNTL experiment to define the location in the SMTH. The fact that there are no "fronts" in the SMTH points to a shortcoming in the methodology being used to identify them. Also, the definition of SST front in the CNTL experiment is also concerning, as it is simply based on an SST gradient. The SST front is not a straight line (I suggest you look at a figure like Figure 1 from Andres (2016)) how do you deal with Gulf Stream rings for example? This also makes me rather skeptical of the definitions of C1, C2 and C3, despite their use in previous manuscripts. Additionally, if there is no defined SST front in the SMTH experiment,**

**then there is minimal relevance to the definitions of C1, C2, and C3 anyway. Andres, M. (2016). On the recent destabilization of the Gulf Stream path downstream of Cape Hatteras. Geophysical Research Letters, 43(18), 9836-9842.**

We thank the reviewer for this comment and for letting us clarify this issue. We did attempt to objectively identify the SST fronts for the SMTH experiments, but this fails because the SST gradient is very weak and homogeneous over a large region such that the detection of the SST front location is mostly determined by numerical noise rather than a defined maximum in the SST gradient. We therefore use the classification based on the CNTL experiments only to be able to compare cyclones within a geographically similar genesis location and track across the experiments. We will adapt the manuscript to point this out more clearly.

Regarding the second point of the reviewer, let us underline that our detection scheme is indeed able detect all lines, including rings, and not only straight lines as the reviewer seems to suspect.

**(M): Line 200 - I cannot understand the logic behind not just looking at all cyclones.**
**Line 223 - you make the bold claim that cyclone intensification is only weakly modified by SST gradient, but what about all those that aren't undergoing maximum intensification right there?**

The Gulf Stream and Kuroshio regions are the areas where the major SST fronts are located and the maximum SST differences after the smoothing are observed. We wanted to relate the intensification of cyclones with the changes in absolute SSTs and/or SST gradient and therefore did not include cyclones with a maximum intensification further away towards the central/eastern parts of the two basins, as the intensification of these cyclones could not be directly associated with these changes, as likely other reasons/mechanisms can evolve.

**(M): Line 258 - This is a main point of the paper and one I don't feel comfortable with. Your claim is that the SST gradient is not particularly important for the intensification of individual cyclones. Whilst I may agree with this general sentiment (see my first major comment), I do not think the analysis presented here can make that conclusion.**
**1) As mentioned previously, you are looking at a subset only.**

We thank the reviewer for sharing these thoughts with us. First of all, please let us clarify that we only used a subset of cyclones (C1,2,3) to investigate the possible changes on cyclone intensification for cyclones belonging in the different categories, for the reasons mentioned in the previous response. For the remainder and main part of the manuscript we used all cyclones. For the climatological analysis we previously used all cyclones in the respective regions (irrespective of track and location of maximum intensification) and in the revised manuscripts we use all cyclones in our database. We will make sure to point out this difference between the parts of the paper more clearly.

**2) There are many differences between your two experiments, not just the SST gradient. The absolute SST changes also, and I strongly suspect that the variability in the Gulf Stream path length/separation from coast changes too, each having been shown to significantly affect the interaction. I don't see anything up to this point that allows you to definitively link it to the SST gradient. You mention in Line 40 that these factors need to be teased apart, but I am not convinced that has been done here simply by defining C1, C2 and C3 - these classifications are also ill-defined in the SMTH experiment.**

On this point we fully agree with the reviewer, and brought up this issue frequently in the discussion of this paper (L25, 37-41, 160-170, 185-187, 244, 250-251, 285-286, 331-334, 350-358, 411-414, 435-437) and our ERA-Interim-based analyses for the same regions (Tsopouridis et al. 2020a,b). Our main conclusion here is that both the absolute SSTs and the SST gradient play roles for the evolution of cyclone characteristics. For instance, the increase (decrease) of the absolute SST values results in

an increase (decrease) of the upward surface heat fluxes and specific humidity, whereas a weak SST gradient in the SMTHG experiment is found to be associated with a considerable reduction in precipitation (in line with Kuwano-Yoshida et al., 2010b), compared to the pronounced SST gradient in the CNTL experiment.

**Line 266- as I alluded to earlier, it does not seem appropriate to use two different subsets of cyclones for each of these sections (individual vs. mean state) if you are going to draw comparisons between them. Additionally, given that the typical timescale for thermal air-sea interaction is ~ a day or so, one would expect the GS/KE to impact some storms further downstream from the region. This again raises questions regarding the conclusion that the SST front does not affect individual cyclones, just because the ones reaching maximum intensity in that specific regions are not that affected.**

Regarding the subsets we refer to our response to L258/point 1.

Regarding the debate on local versus remote effects, we appreciate that our previous climatological analysis was not ideal to comment on this issue, but the revised analysis will allow us to comment on this issue. Counting all air-sea exchange within different radii of a cyclone centre as "local", we will show that "non-local" air-sea exchange dominates the climatology. This finding is in line with previous studies that tried to clarify the influence of surface heat fluxes on midlatitude cyclone development and untie their local and remote effects (e.g., Nuss and Anthes 1987; Kuo and Reed 1988; Haualand and Spengler 2020). We thus believe that SSTs and SST fronts are mainly important for climatologically setting the environment in which cyclones evolve, even though each individual cyclone is not significantly affected directly by changes in the SST.

Regarding the "variability in the Gulf Stream path length/separation from coast", the heavy smoothing in the SMTHG experiment implies that there is actually not even a "Gulf Stream" anymore, not only that the length has changed.

**(M): Section 3.4. Line 294 -Parfitt and Czaja (2016) have shown that the top 30% of climatological latent heat fluxes in the Gulf Stream region are associated with cyclones that have just passed over the region. It is continuously said here "when no cyclones are present in the respective region", however that simply means the cyclone center is not directly above the region - this does not mean a cyclone's passing isn't responsible for the trailing cold air that maximises the fluxes. This is relevant for consistency with Section 3.4.2 later, where it is shown cyclone precipitation does change significantly. A lot of moisture availability, that is ultimately taken up in say the warm conveyor belt of any given cyclone, in these regions can come from the strong latent heat fluxes**
**behind the cold front of a system that passed ahead of it. This comment is also relevant for your statement "for specific humidity, cyclones account only for a small part of the climatological differences" in Line 359, and for your statement from Line 379-383. In particular, I do not agree with Line 381-383. This comment feeds back to the previous one, where I do not think conclusions about the SST influence on the atmosphere through cyclones can be drawn simply by comparing whether a cyclone is in that specific region or not.**

We acknowledge the reviewer's concerns and agree that the attribution to only cyclone centres in the box might have been too restrictive. Please refer to our response in the first comment on the additional analysis we will include in the revised version of the manuscript.

---

## Author Response (AR1)

**Response to Reviewer 1**

We sincerely thank Irina Rudeva for her constructive critique and detailed feedback of our manuscript. She addressed important and valid points that helped us to improve the manuscript. We provide detailed responses with our comments and changes in blue.

**MAJOR COMMENTS**

**1. The finding that cyclones play a secondary role in forming strong sensible and latent heat fluxes is only true for the low-pressure systems which have their centres in the selected areas, i.e. right over the oceanic currents. However, anomalous turbulent fluxes may not be associated with the cyclones with centres in the selected boxes. Tilinina et al. (2018; https://doi.org/10.1175/MWR-D-17-0291.1) showed that extreme fluxes over the Gulf Stream are linked to regions of cyclone-anticyclone interaction (usually associated with strong winds and large air temperature anomalies) when they are located above the ocean current. The finding by Tilinina et al. implies that the atmospheric circulation plays an important role in creating strong turbulent fluxes. While systems that have their centres within the selected boxes may, indeed, not be associated with strong air-sea fluxes, I believe that the paper should focus on a different subset of cyclones, i.e., those that are located ~ 1000-3000 km to the north or north-east of the Gulf Stream or Kuroshio). This is in agreement with the area of the largest changes of cyclone activity shown in Fig.4.**

We thank the reviewer for this comment and agree with her that the attribution to only cyclones in the box was indeed restrictive. We thus complemented our previous analysis with one in which we considered the pertinent variables occurring within a radius of 750 km (consistent with a maximum cyclone circumference in Wernli and Schwierz (2006) and the analysis of Rudeva and Gulev (2011)) of any cyclone over its entire lifetime in the ocean basin, irrespective of its propagation and its location of maximum intensification. We adjusted our manuscript and present the new results in the following Sections: Introduction (lines: 96-99), Data & Methods (lines: 165-171), in subsections 3.4.1 (lines: 316-319), 3.4.2 (lines: 372-375), 3.4.3 (lines: 386-401), 3.4.4 (lines: 417-423), and in the Conclusions (lines: 456-470). We also changed figures 7-10 (c-d, g-h) and their associated captions, respectively. The results arising from this new analysis are in line with our previous results and support our previous conclusions. The only exception is Figure 10 (g-h), in which we now find the wind speed composite at 300 hPa that is not associated with cyclones to better resemble the climatological differences in the North Pacific, which is now in line with the results for the North Atlantic. In addition, we conducted our analysis within different radii around the cyclone centers (500 and 1000km) and present the results for the surface heat fluxes and the different types of precipitation in Appendix B (Figures B1-B4).

**2. I do not feel comfortable with the idea of using SST fronts from CNTRL for analysis of the smoothed runs. I believe that fronts should be objectively identified in all runs. This is particularly important in the Pacific, where SST fronts can hardly be seen in the SMTHK experiments. As the classification of cyclones in this paper is based on the location relative to the SST front, in those cases when an SST front in SMTH cannot be detected, the type of cyclones cannot be defined. That said, given my first comment on the subset of cyclones that may be particularly affected by the SST gradient, I am not sure that a classification based on the location of cyclone centres relative to the ocean current is particularly important. Fig. 5 shows that there are hardly any statistically significant differences between those types, not to mention the differences between SMTH and CNTRL runs.**

We thank the reviewer for her comment that our rationale for this choice has not become apparent in the previous version of the manuscript. We use this classification only to be able to compare cyclones with a geographically similar genesis location and track across the experiments. We adapted the manuscript accordingly to point this out more clearly (please refer to subsection 2.4 (lines: 158-163)). Having said this, we did attempt to objectively identify the SST fronts for the SMTH experiments, but this fails because the SST gradient is very homogeneous over a large region such that the detection of the SST front location is mostly determined by numerical noise rather than a defined maximum in the SST gradient.

**3. Finally, it will be interesting to assess how much changes in heat fluxes, precipitation, etc. scale with a change in the SST gradient. It is mentioned multiple times that smaller reduction in the SST gradient in the Kuroshio region led to smaller corresponding changes in the atmospheric circulation. The question I want to raise here is if similar reductions in the SST gradient in those basins lead to comparable response. Or, alternatively, how stronger/weaker smoothing affects the same region.**

We thank the reviewer for raising this issue. As mentioned in the "Data and Methods" section, we use data from model experiments that were provided by JAMSTEC. For this reason, we unfortunately do not have the capability to run additional experiments to estimate how stronger/weaker smoothing affects the same region (the Gulf Stream or the Kuroshio region). Nonetheless, we thank the reviewer for sharing with us this interesting idea, which is an interesting potential analysis in the future, when we will be capable to execute more/different experiments.

**OTHER COMMENTS**

We thank the reviewer for all the "other/minor comments" she raised, which we considered when editing the manuscript. Our responses to each comment are following:

**Title: In the current version of the manuscript, it is, indeed, shown that the atmospheric conditions change stronger in the absence of cyclones over the ocean currents. However, as I said in the above, if more analysis is done on cyclones in the storm track areas, the title may need to be changed.**

With the new analysis considering the entire ocean basins, we confirmed our previous findings and thus the choice of the title. We just changed the order of some words in the title, between the previously submitted and revised version of the manuscript. The current title is: "The Gulf Stream and Kuroshio SST fronts affect the winter climatology primarily in the absence of cyclones".

**l.90: Instead of just saying 'to the questions raised above', it is worth repeating the main questions here.**

We thank the reviewer for this comment and rephrased the respective sentence (please refer to lines 99-100).

**Data: I could not find the time resolution of the data used.**

We thank the reviewer for highlighting this missing information. We analyse the AFES output on a 0.5° horizontal grid at 6 hourly intervals. We added this information in subsection 2.1 (line: 109).

**l.97: Why the period of integration ends in 2001? Considering the coarse SST resolution in ERA-Interim prior to 2002, selection of the time period needs to be justified.**

As indicated previously (please see our response to your Major comment 3), the model experiments we use were conducted for the period 1982-2001 and this is why we are not able to extend this time step after 2002. But this is not a caveat, as the AFES simulations do not suffer from the same limitation as ERA-Interim, but use high-resolution SSTs as boundary conditions for the entire time period.

**l.104: Explain what is meant by "1-2-1 running mean"**

It is a three-point filter with the weights 0.25, 0.5 and 0.25. The 1-2-1 filter has a sharp cutoff, so that unwanted frequency components are effectively removed. This filter was applied to the data prior to JAMSTEC providing it to us. For further information please see Kuwano-Yoshida and Minobe, 2017 (https://journals.ametsoc.org/view/journals/clim/30/3/jcli-d-16-0331.1.xml?tab_body=fulltext-display). We added the additional information in the revised version of the manuscript (please refer to subsection 2.1, lines: 115-117)

**l.114: Give a more detailed description of how the jet was detected.**

We thank the reviewer for letting us clarify this issue. To assess the potential impact of the SST smoothing on the upper levels, we detect the position of jet following the algorithm of Spensberger et al. (2017). The algorithm detects jets by their axis, the line of maximum winds that separates cyclonic from anticyclonic wind shear. The algorithm requires the wind maximum to be well-defined, but does not impose a strict minimum wind speed (Spensberger et al., 2017). We added this information in subsection 2.2 (lines: 131-134).

**l. 123: following a comment above, 3 consecutive time steps for the minimum life time of 12 hr comes from nowhere. I'd mention here that 12 hours is less than more often used threshold of 24 hours (i.e., five 6-hour time steps), applied in Neu et al. (2013; https://doi.org/10.1175/BAMS-D-11-00154.1) and adopted in many recent studies on extratropical cyclones.**

We thank the reviewer for letting us clarify this issue. Following Neu et al. (2013), we indeed followed the same and often used technique, considering cyclone tracks with at least five 6-hour time steps, but in addition we required cyclones to have three 6-hour time steps in the Gulf Stream or the Kuroshio region in order to classify the cyclones into different categories (C1-5). Asking for five 6-hour time steps only in the "box"/area of interest would significantly reduce the total number of tracks, as we are particularly interested (and applied the respective criterion in the detection algorithm) in "strong" cyclones, including rapidly developing and propagating systems. Given the geographically restricted region and considering that the specific areas are regions of cyclogenesis, a further restriction would downgrade the significance of our findings. We added this missing information in the manuscript, in subsection 2.3 (lines: 143-144) and in subsection 2.4 (lines: 152-154).

**l.124, 144: I presume that you require that the cyclone centre, not just any point of the cyclone area, passes over the Gulf Stream or Kuroshio regions?**

The reviewer suspected correctly. However, this method is not valid anymore, due to the new analysis we conducted (as described in detail to our response in Major Comment 1).

**l.159: Also refer to Fig. 2a,b for the SST gradient.**

We thank the reviewer and added this missing information in subsection 3.1 (lines: 186-187)

**Fig. 3: what are the units used to show the distribution of the SST fronts and jet axis? The caption says 'km of line/ 100km**2'. It suggests that the SST front is represented by a line, not**

by an area where the SST is above a 2 or 1.25K/100km (this indicates again, that the method used to define the SST front needs to be better described in Section 2.2, see my earlier comment). Explain how you got km of line per a unit area. Why not showing the frequency of the SST values above a threshold instead? As I do not quite understand the units in Fig.3, I am not sure how to interpret higher jet distribution values over the Pacific.

The reviewer is right. We do detect SST front lines rather than regions of strong SST gradients. We require front lines in order to be able to define when cyclones cross the front. We pointed this out more clearly in subsection 2.2 (lines: 126-127).
For the climatologies and composites we normalise the occurrence of both SST front lines and jet axis lines to account for the varying area covered by grid cells. We achieve this by showing the average length of SST front line/jet axis line per unit area, hence the resulting unit of length per area. For details on the normalisation, we refer to Spensberger and Spengler (2020). To compile a climatology, we use for both the SST front and the jet axis distributions a unit of length per area, due to its independence of the grid resolution. We added this information in subsection 2.2 (lines: 135-138).

**"The text says that the jet is 'stronger', does that mean that it is present more often or that it is wavier?"**

If the density is more widespread, the jet is wavier or shifts more in latitude. If the density is focused, the jet is usually mostly at this location. Woollings et al. (2018) found that a more focused jet is usually also associated with the jet being stronger. Hence, we use more focused synonymously with stronger.

**l. 172: How do you know that it is the jet that affects the cyclones, not the other way round? More importantly, how does this section answers the question raised at the start of the paragraph (on the impact of the upper-level circulation on cyclones)? I only see a comparison of statistics at the upper and low levels.**

We thank the reviewer for this comment. The importance of the jet for intensifying cyclones has been confirmed in numerous studies (e.g., Evans et al., 1994; Schultz et al., 1998; Riviere and Joly, 2006; Tsopouridis et al., 2020b). Please refer to the Introduction Section (lines:42-48). Nevertheless, we acknowledge that the topic sentence of this paragraph was confusing and rephrased accordingly to better express the content of the paragraph, that is the climatological position of the jet stream over the two ocean basin, with respect to the position of the respective SST fronts in the two regions. Please refer to subsection 3.2 (lines: 199-200) for our change.

**l. 179-187: As the Pacific jet is located further south compared to the Atlantic jet, it is hard to shift it more equatorward.**

We agree with the reviewer and only document the observed differences (lines: 208-209).

**l. 188-191: Again, I do not see larger changes in the cyclone activity in the Atlantic. The patterns in the Atlantic and Pacific are slightly different, but this may be a result of the different location of the SST front/jet axis, as well as difference in the shape of continents. Moreover, I see a shifted storm track activity in the Atlantic or Pacific, however, basin-wide it does not seem to be weaker as opposed to the Southern Ocean mentioned in the text.**

We explicitly refer here to the Gulf Stream and Kuroshio regions (as these are indicated by a black box, respectively) and not for the Atlantic and Pacific basin as a whole. In the Gulf Stream region there is indeed a larger reduction in storm track activity ($2*10^{-6}$ km$^{-2}$), compared to the Kuroshio

region (maximum of $1*10^{-6}$ km$^{-2}$). Given the fact that the SST front (despite significantly weaker in the SMTH experiments) is located at the same regions after the SST smoothing (please refer to Figure 2) and the shape of the continents is unchanged between the experiments, we relate this difference to the sharper SST smoothing, and thus a considerably weaker SST gradient in the Gulf Stream region (please compare Figure 2c with Figure 2f).

**l.192: The differences in the Atlantic are also shifted between the upper and lower levels, less than in the Pacific, but in the same direction. It will be interesting to add cyclone frequencies in the middle of the troposphere (e.g., 500hPa).**

We thank the reviewer for her comment. We agree with the reviewer that the differences in the Atlantic, between the upper and lower levels, are shifted in the same direction and refer to this on lines: 205-208. An additional analysis of cyclone frequencies in the mid-troposphere is beyond the scope of our study.

**Fig. 5: If you keep this plot in the revised manuscript, could you estimate if the difference between lines (i.e., between mean values for the different cyclone types at every time step) are statistically significant and mark the time steps when the mean value for a given type is significantly different from the other two types?**

We thank the reviewer for this comment. As the reviewer can see in Figure 5a,b, the results are statistically significant around maximum intensification (from -6h to 6h) for C2, where the 50$^{th}$ percentile of C2 coincides with the 75$^{th}$ percentile of C1 and with the 75-100$^{th}$ percentile (not shown in colors) of C3. Moreover, we mark that the median (50$^{th}$ percentile) of C2 is always above the ones for C1,3 during the 48h period in both experiments for the Gulf Stream region, indicating a clear tendency for higher intensification in C1,3, compared to C2. In the Kuroshio region (please refer to Figure 5d,e) there is a lot of overlap in the Interquartile Range (IQR) for the 3 categories and a larger variability. Nevertheless, again the median of C2 is above the ones for C1,3 before the time of maximum intensification, indicating a clear tendency for weaker intensification in C2, compared to C1,3. We thank once more the reviewer for pointing out this missing information and we added a discussion in subsection 3.3 (lines: 240-244).

**l.250, Fig.6: I am not sure if Fig 6 is worth showing. As the cyclone classification in this paper is based on the location relative to the SST front, the mean SST in the cyclone area is somewhat prescribed.**

As the reviewer correctly indicates, the cyclone categorisation is based on the location of cyclones relative to the SST front. However, as we explicitly refer to in lines 190-191, it is not only the SST gradient that changes between the experiments, but also the absolute SST. Moreover, the absolute SSTs are important independently, or in conjunction with the SST gradient, for the intensification of cyclones in the two regions, which we discuss in more detail in the Introduction Section (please refer to lines: 24-41). We thus decided to keep Figure 6, in which the absolute SST changes between the experiments for the different cases and regions are captured in detail and can be related with the intensification of cyclones (Figure 5).

**l.256: In line with my earlier comments, the statement that weaker SST gradients in the Pacific are not strong enough to affect cyclone development, points to the fact that, perhaps, the SST threshold for the Pacific regions should be increased. Ideally, I think, they should be identical (even if the Atlantic threshold is decreased).**

We thoroughly tested different thresholds to obtain reasonable SST fronts in both basins. Given the different SST gradient in the two basins (e.g., Nakamura et al., 2004; Tsopouridis et al., 2020b), if

using the Atlantic threshold, we would capture almost no SST fronts in the Kuroshio region. Vice versa, with the Pacific threshold in the Gulf Stream region we would obtain many weak SST fronts that would make it difficult to identify the main SST front. We thus believe that a different threshold is necessary to accommodate the different natures of the boundary currents and SST fronts. To clarify the use of different thresholds in the two ocean basins, we adapted the text accordingly in subsection 2.2 (lines: 127-130).

**l.303: As I said in the major comments, cyclones that have centres over the oceanic currents are not expected to cause strong heat fluxes.**

Following the concern expressed from the reviewer in her Major Comment 1, we conducted a new analysis and found that the result is valid, not only for cyclones that have centres over the oceanic currents.

**l. 307, Section 3.4.2: In line with my Major comment 1, it will be interesting to show how much precipitation changes with cyclones most affected by the SST change. (however, the increased precipitation over the ocean currents may be associated with another subset of cyclones, not those that are related to the increased heat fluxes).**

Following the concern of the reviewer expressed in Major comment 1, we now include all cyclones, not only the ones with a maximum intensification in the GS/Kuroshio region and can estimate the overall contribution of cyclones after the SST smoothing in the two basins. For more details, please refer to our response in Major comment 1.

**l.313: Does smoothed SST in the Atlantic lead to a reduced separation distance between the maximum in large-scale and convective precipitation? If that does not happen, then explain why smaller SST gradients in the Pacific are responsible for more collocated precipitation maxima in the Pacific.**

We thank the reviewer for her suggestion, which helped us clarify this issue. Smoothed SST in the Atlantic does not lead to a reduced distance between the maximum in large-scale and convective precipitation. We thus deleted the hypothesis raised in the previous version of the manuscript. Please refer to lines 339-340 in the revised version of the manuscript.

**l.323: the largest shift in cyclones occurs downstream of the selected regions, however, the precipitation changes most significantly within the black boxes. I think it should be discussed in the text.**

We thank the reviewer for her comment. In the revised Figure 8 c,d, g,h, we show that a considerable fraction of the precipitation differences after the SST smoothing is not associated with cyclones. Our results are thus consistent with what the reviewer indicates, that is: the more significant precipitation change occurs in the western parts of the two ocean basins, where the largest changes in SST are observed. We discuss this in subsection 3.4.2 (lines: 357-362).

**l.349: specific humidity can be affected by cyclones over the central part of the North Atlantic. Besides local change in evaporation over the ocean currents, changes in characteristics of the extratropical cyclones downstream of the Gulf Stream and Kuroshio may modify the intensity of warm conveyor belts. There is also an increase in the relative humidity in the south-east corner of the Pacific basin, most likely associated with a southward shift of the storm track.**

We thank the reviewer for this comment. Overall, we found the specific humidity, which is not associated with cyclones (outside of a 750 km-radius) to better resemble the climatological

differences after the SST smoothing. Nonetheless, when considering the specific humidity that is associated with cyclones (within a 750 km-radius), we indeed observe an increase in the central Atlantic and relate this with a shift to the southeast of the storm track in the SMTHG experiment. Please refer to subsection 3.4.3 (lines: 389-391).

**l.395: The pattern in the Atlantic (jet, cyclone frequency) suggests negative NAO in the smoothed runs**

We want to the thank the reviewer for sharing her thoughts with us. Regarding the NAO phase, we avoided such an association, as this would need a further analysis to be confirmed, and this is beyond the scope of this study.

**l.400: I like the hypothesis here. However, in my opinion, the jet response in the Pacific is mainly associated with a strong increase in cyclone frequency in the Bay of Alaska (Fig.4c). A well developed cyclonic circulation will increase (decrease) the wind speed to the south(north) of its centre. This low pressure system may also be responsible for the moisture advection from the central Pacific to north-east (Fig. 9f). Interestingly, cyclones over the Kuroshio are associated with an intensified subtropical jet.**

We thank the reviewer for her comment. Our results based on the new analysis indicate that the wind speed difference that is not-associated with cyclones (outside of the 750km-radius, Fig. 10d,f) resembles more the difference of the wind speed climatology (Fig.10 b,f) in both regions, compared to the upper-level wind speed difference that is associated with cyclones (Fig.10 c,g). We relate our previous findings to the restriction of the cyclones' subset used before and are confident that our new analysis (based on the reviewer's major comment 1) more completely interprets the upper-level wind speed field.

**Fig. 7-10: Indicated statistical significance of the differences.**

We thank the reviewer for raising this issue. In the revised version of the manuscript, we now provide the statistically significant results (>95%, based on a t-test) and present the new results in Figures 7-10 (b,f), as the reviewer suggested.

**MINOR COMMENTS**

**l.6: I'd replace the 'intensification' with 'intensity', unless you really want to stress the process of deepening, rather than the maximum strength of the systems.**

As the reviewer suspected we are particularly interested in the intensification of cyclones, rather than their intensity. We thus kept the word "intensification" in this sentence as it reflects what we want to stress.

**l.14: change 'states' to 'state' &**
**p.1, l17: and throughout the paper: change to 'regions' as the Gulf Stream and Kuroshio are two independent regions (as opposed to, e.g., the Kara and Barents Sea) &**
**l376: change to '40%'**

We thank the reviewer for pointing out this spelling errors. We corrected them in the revised version of the manuscript.

**l.102: Introducing SMTHG and SMTHK, it is worthwhile mentioning the Gulf Stream and Kuroshio, respectively, instead of the North Atlantic and Pacific. Otherwise, it is not immediately obvious what K and G stand for.**

We agree with the reviewer, but replaced it with "the greater area around the Gulf Stream" and "the greater area around the Kuroshio Extension" to avoid confusion with the terms "Gulf Stream region" and "Kuroshio region", as these are used in the manuscript. Please, refer to lines: 113-114 for the respective change.

**l.145: replace "intensification" with "intensity".**

The word "intensification" has been correctly used and is thus kept in the revised version. We made this choice to highlight the difference with the set of cyclones used for the categorisation/classification (as described in the subsection 2.4). For the different categories (C1-5) we indeed considered cyclones with a maximum "intensification" (and not "intensity") in the Gulf Stream and Kuroshio regions.

**l.199: Perhaps, 'classification' sounds better.**

We agree with the reviewer's suggestion and replaced "categorization" with "classification".

**Fig. 1, caption: Remove the first '(a)' from the caption. In c (and the following figures), make clear that it is SMTHG minus CNTL. The caption should mention black solid lines.**

We thank the reviewer for the comments and proceeded with the respective correction and edits in the captions.

**Fig. A1, panels b and e: The range of values in the colour scale is too big and yellow shading is hard to see against the white background. The colour scale does not match either Fig. 2a or 2d, so I do not see why you chose such large range.**

We understand the concern of the reviewer (that the range in Figure A1 is large), however as we want to compare the respective results between the model (AFES) and Reanalysis (Era-Interim) we believe that using the same range and colour scale is useful as it will be more straightforward for the reader to capture the differences between the two datasets by using the same range.

**Fig.2, b, e:  Why did you choose to show the SST gradient in 0.5K/100km instead of K/100km?**

We thank the reviewer for letting us clarify this issue. As the reviewer realizes herself  in the previous comment ("yellow shading is hard to see against the white background"), using K/100km instead would result in an almost white background, given the significant smoothing of the SST. We decided thus instead to show the SST gradient in 0.5K/100km and highlighted the different units used between panels a,d and b,e in both the figure and the caption.

**Fig.7-10: I do not think I understand the last sentence in the captions. What kind of scaling was applied? In this case please describe in the Methods.**

We thank the reviewer for the comment. After applying a different analysis, the sentence in the caption is removed.

**Fig. 3: On my screen the 'orange' colour scale looks nowhere near to orange, I'd call it pale red. Perhaps, change the scale a little bit.**

Following the reviewer's suggestion, we replaced the word "orange" with "pale red" in the caption.

**References**

Evans, M. S., Keyser, D., Bosart, L. F., & Lackmann, G. M. (1994). A satellite-derived classification scheme for rapid maritime cyclogenesis. Monthly weather review, 122(7), 1381-1416.

Kuwano-Yoshida, A., & Minobe, S. (2017). Storm-track response to SST fronts in the northwestern Pacific region in an AGCM. Journal of Climate, 30(3), 1081-1102.

Nakamura, H., Sampe, T., Tanimoto, Y., & Shimpo, A. (2004). Observed associations among storm tracks, jet streams and midlatitude oceanic fronts. Earth's Climate: The Ocean–Atmosphere Interaction, Geophys. Monogr, 147, 329-345.

Rivière, G., & Joly, A. (2006). Role of the low-frequency deformation field on the explosive growth of extratropical cyclones at the jet exit. Part II: Baroclinic critical region. Journal of the atmospheric sciences, 63(8), 1982-1995.

Rudeva, I., & Gulev, S. K. (2011). Composite analysis of North Atlantic extratropical cyclones in NCEP–NCAR reanalysis data. Monthly weather review, 139(5), 1419-1446.

Schultz, D. M., Keyser, D., & Bosart, L. F. (1998). The effect of large-scale flow on low-level frontal structure and evolution in midlatitude cyclones. Monthly weather review, 126(7), 1767-1791.

Tsopouridis, L., Spensberger, C., & Spengler, T. (2020). Characteristics of cyclones following different pathways in the Gulf Stream region. Quarterly Journal of the Royal Meteorological Society.

Tsopouridis, L., Spensberger, C., & Spengler, T. (2020). Cyclone intensification in the Kuroshio region and its relation to the sea surface temperature front and upper-level forcing. Quarterly Journal of the Royal Meteorological Society.

Wernli, H., & Schwierz, C. (2006). Surface cyclones in the ERA-40 dataset (1958–2001). Part I: Novel identification method and global climatology. Journal of the atmospheric sciences, 63(10), 2486-2507.

Woollings, T., Barnes, E., Hoskins, B., Kwon, Y.-O., Lee, R.W., Li, C., Madonna, E., McGraw, M., Parker, T., Rodrigues, R., Spensberger, C., Williams, K. (2018). Daily to decadal modulation of jet variability. Journal of Climate, 31, 1297-1314

**Response to Reviewer 2**

We sincerely thank the anonymous reviewer for his/her feedback and comments, which helped us to improve the manuscript. We provide detailed responses with our comments and changes in blue.

**(M): The introduction provides a good comprehensive overview of many aspects of GS/KE - atmosphere interaction, but fails to discuss a large body of work directly relevant to the main focus of this paper i.e. SST influence on time-mean as affected by changes in synoptic storms. The following studies have already considered contributions of storms to the mean atmospheric state in the Gulf Stream and Kuroshio regions (Parfitt and Czaja, O'Neill et al, Parfitt and Seo, Masunaga et al, a,b). The first two of these studies show that the mean state in the tropospheric wind fields and precipitation are set by extreme synoptic situations, with the latter four studies specifically highlighting that it is the atmospheric fronts (and not really the cyclones). Indeed, when you remove the atmospheric fronts from the climatology, you remove the time-mean convergence signal above the Gulf Stream and Kuroshio altogether (see Figure 2 in Parfitt and Seo) as the remaining baroclinic components mostly cancel out (see Figure 3 and Supplementary Figure 2 in Parfitt and Seo). Furthermore, atmospheric fronts contribute ~90% of the rainfall in these regions (Catto et al., 2016).**
**One of your main conclusions (that changes in individual cyclones resulting from your SST changes aren't important for changes in the mean state) is therefore not surprising given the results of these aforementioned studies (that show the mean state in both winds and precipitation is set by atmospheric fronts). I think the authors should discuss these studies, as well as the consistency of their results with them. Furthermore, I think it is fair (and scientifically interesting) to request that the authors make some attempt to use these studies together with their results to make some additional hypothesis regarding the relationship between SST front, cyclone and atmospheric front (e.g. SST frontal influence on mean-state mostly through weak atmospheric fronts/weak cyclones, as discussed in Masunaga et al, 2020b, or some other idea).**

We thank the reviewer for this comment, and we included these papers in the Introduction Section (lines: 73-82) of the revised manuscript. In particular the findings of Masunaga et al., 2020a,b are very much in line with our findings as well as the findings in a related submitted study focusing on the interaction of atmospheric fronts with SST gradients (Reeder et al., resubmitted to JAS). We would, however, caution whether what Masunaga et al., 2020a,b refer to as "weak systems" is representative of cyclones. We thus complement the discussion presented in Masunaga et al. 2020a,b with our cyclone-centred viewpoint.
As fronts are specifically discussed in a related submission (Reeder et al., resubmitted to JAS), we decided to not duplicate this analysis and discussion here.
Following the concern from the reviewer, we however complemented our previous analysis with one in which we considered climatologies of the pertinent variables occurring within a radius of 750 km (consistent with a maximum cyclone circumference in Wernli and Schwierz (2006) and the analysis of Rudeva and Gulev (2011)) of any cyclone over its entire lifetime in the ocean basin, irrespective of its propagation and its location of maximum intensification. We adjusted our text and present the new results on the following Sections Introduction (lines: 96-99), Data & Methods (lines: 165-171), in subsections 3.4.1 (lines: 316-319), 3.4.2 (lines: 372-375), 3.4.3 (lines: 386-401), 3.4.4 (lines: 417-423), and in the Conclusions (lines: 456-470). We also changed figures 7-10 (c-d, g-h) and their associated captions, respectively. The results arising from this new analysis are in line with our previous results and support our previous conclusions. The only exception is Figure 10 (g-h), in which we now find the wind speed composite at 300 hPa that is not associated with cyclones to better resemble the climatological differences in the North Pacific, which is in line with the results for the North Atlantic. In addition, we conducted our analysis within different radii (500 and 1000km) and present the results for the surface heat fluxes and the different types of precipitation in Appendix B

(Figures B1-B4). With the larger radius, we also included the contribution of nearly all fronts in the analysis and thereby implicitly addressed the reviewers concerns about the role of atmospheric fronts.

**Line 80: Following on from the above, the strong surface fluxes primarily occur behind cold fronts that bring cold dry air off the continent, not necessarily with cyclones as a whole (especially when the cyclones are positioned directly over the GS), and are modulated thus. Also, this paragraph should clearly reference Bishop et al. (2017) which discusses the scale dependence of SSTs and heat fluxes in the western boundary current regions in terms of atmosphere-driven vs. ocean-driven.**
**Bishop, S. P., Small, R. J., Bryan, F. O., & Tomas, R. A. (2017). Scale dependence of midlatitude air–sea interaction. Journal of Climate, 30(20), 8207-8221**

Following our response to the previous comment, we believe that a large enough radius would include a significant part of the cold sector associated with the parent cyclone. However, cold air outbreaks must not necessarily occur in the cold sector of a cyclone, but can also be associated with an anticyclone (e.g., Dalavalle and Bosart 1975; Colucci and Davenport 1987; Nakamura et al., 2016). Hence, climatological features associated with these type of cold air outbreaks should not be accounted for in a cyclone specific climatology.
We thank the reviewer for pointing us to the paper by Bishop et al. (2017). We acknowledge this debate on whether the SST variability in the WBCs is atmospheric or ocean-driven on different timescales. However, with our analysis based on simulations with prescribed SSTs we feel that we cannot contribute to this debate and thus decided to not discuss our results in the context of Bishop et al. (2017) in the revised version of our manuscript.

**(M): I have a few serious concerns about the data used for this study. Firstly, it has been shown many times how important model resolution is for accurately resolving air-sea interaction in western boundary current regions. Some examples: Smirnov et al. (2015) show the necessity for both ocean and atmosphere at 0.25deg in order for SST-induced heating to be balanced by transients rather than cold-air advection. At 50km, you are not there, and barely resolving the cross-frontal scale in the atmosphere (~100km). It is also well known that you need eddy-resolving (~1/10deg) resolution in the ocean to actually resolve many of the oceanic eddy processes crucial to the GS/KE air-sea interaction (e.g. Figure 2 from Hewitt et al., 2017). These eddies are known to be crucial for the interaction (Ma et al., 2015) and its influence on general atmospheric and oceanic variability. Your resolution falls under these values and I don't see how definitive conclusions can be made given we already know the interaction cannot be fully resolved. The importance of resolution and resolving transients for the hemispheric flow is also shown in Lee et al. (2018). I also can't understand why ERA-Interim is used instead of ERA-5 which is available, which has a much better resolution atmosphere and ocean. You even note this yourself in Line 157. Please also mention in Section 2.1 what temporal resolution you use. How often do you calculate SST fronts etc?**

Regarding the first point of the reviewer on the model resolution, we would like to make three remarks. First, the AFES simulations on which we based our analysis have been used for numerous studies on the effect of SST gradients and fronts on the atmosphere (e.g., Nakamura et al., 2008; Kuwano-Yoshida et al., 2010; Kuwano Yoshida and Minobe 2017). We thus regard the AFES data as a valid dataset in the context of previous studies on this topic.
Second, we appreciate and are aware that smaller-scale ocean features can affect the climatological air-sea exchange to some extent. However, as we only focus on 1st-order effects on synoptic-scale systems, it seems implausible that ocean eddies will fundamentally change these attributions and thus our conclusions.
Finally, our analysis based on this coarser model data is highly relevant to the climate modelling community.

Regarding ERA-Interim, we included this comparison to the AFES climatology only to make plausible that the AFES simulations, which are comparable in resolution to ERA-Interim, provide a reasonable depiction of the air-sea exchange and storm track over our target region. Moreover, we have tested the sensitivity of our results to either just using 1979-2001 or 2002-2016, with no significant changes between the two periods despite the resolution change in SST for ERA-Interim (e.g., Masunaga et al. 2015). Overall, ERA-Interim has been extensively and fruitfully used in numerous climatological analyses of air-sea interactions. We thus argue that it remains a relevant data set.

Finally, regarding the SST fronts, we perform the detection using SST data filtered with a spectral truncation to T84 resolution and detect the SST fronts in the instantaneous SST field every 6-hours. We added the missing information in the revised version of the manuscript (please refer to subsection 2.2, lines: 125-126).

**Line 110 - What is the sensitivity of your results to choice of threshold on the SST gradient? Also, how can you confidently make comparisons between the Atlantic and the Pacific when you have a different threshold for both?**

We thoroughly tested different thresholds to obtain reasonable SST fronts in both basins. Given the different SST gradient in the two basins (e.g., Nakamura et al., 2004; Tsopouridis et al., 2020b), using the Atlantic threshold, we would capture almost no SST fronts in the Kuroshio region. Vice versa, with the Pacific threshold in the Gulf Stream region, we would have obtained many weak SST fronts that would make it difficult to identify the main SST front. We thus believe that a different threshold is necessary to accommodate the different natures of the boundary currents and SST fronts. We adapted the text accordingly (subsection 2.2 (lines: 127-130)), to clarify the use of different thresholds in the two ocean basins.

**Line 120 - What about the sensitivity to cyclone detection? Neu et al. clear show vastly different climatologies depending on the specific method. Neu, U., Akperov, M. G., Bellenbaum, N., Benestad, R., Blender, R., Caballero, R., ... & Grieger, J. (2013). IMILAST: A community effort to intercompare extratropical cyclone detection and tracking algorithms. Bulletin of the American Meteorological Society, 94(4), 529-547**

We thank the reviewer for raising this issue and letting us discuss it in more detail this issue. We conducted several sensitivity experiments where we varied parameters in the detection and tracking namelists used by the algorithm. As the reviewer mentioned, there are qualitative differences between previous studies, but unfortunately no sufficient information is provided regarding the parameters used in each study. Motivated by this gap, we decided to publish our detection and tracking namelist in Tsopouridis et al., 2020a. The cyclone density pattern is in good spatial agreement with previous studies (e.g., Hanley and Caballero, 2012; Neu et al., 2013), successfully capturing the major regions of cyclone activity in both basins. We observe small quantitative differences compared to the density climatology presented by both Neu et al. (2013) or Murray and Simmonds (1991a), who also used the Melbourne University algorithm. These small deviations are most likely due to the neglect of shallow and weak systems in our database. We thus included the relevant discussion in the subsection 2.3 (lines: 145-149).

**Line 123-125 - This seems like a strange and unnecessary choice to me, given the importance of quasi-stationary systems in the region (Masunaga et al., 2020)**

We only consider actively developing cyclones in the region and further investigate their evolution and the associated mechanisms. Moreover, the steep orography around Greenland's coast would result in artefacts in the detection and therefore we do not include quasi-stationary systems.

**(M): Section 2.4. I see this as a serious concern. Firstly, I have a problem with using the SST front definition from the CNTL experiment to define the location in the SMTH. The fact that there are no "fronts" in the SMTH points to a shortcoming in the methodology being used to identify them. Also, the definition of SST front in the CNTL experiment is also concerning, as it is simply based on an SST gradient. The SST front is not a straight line (I suggest you look at a figure like Figure 1 from Andres (2016)) how do you deal with Gulf Stream rings for example? This also makes me rather skeptical of the definitions of C1, C2 and C3, despite their use in previous manuscripts. Additionally, if there is no defined SST front in the SMTH experiment, then there is minimal relevance to the definitions of C1, C2, and C3 anyway. Andres, M. (2016). On the recent destabilization of the Gulf Stream path downstream of Cape Hatteras. Geophysical Research Letters, 43(18), 9836-9842.**

We thank the reviewer for this comment and for letting us clarify this issue. We did attempt to objectively identify the SST fronts for the SMTH experiments, but this fails because the SST gradient is very weak and homogeneous over a large region such that the detection of the SST front location is mostly determined by numerical noise rather than a defined maximum in the SST gradient. We therefore used the classification based on the CNTL experiments only to be able to compare cyclones within a geographically similar genesis location and track across the experiments. To point this out more clearly, we adjusted the text accordingly (please refer to section 2.4 (lines: 158-163)).

Regarding the second point of the reviewer, our detection scheme is indeed able to detect all lines, including rings, and not only straight lines as the reviewer seemed to suspect.

**Figure 3: It is not clear what I am looking at here-also, how do you arrive at the units? Also, please provide further information on the jet detection method**

We thank the reviewer for letting us provide more information on this. To assess the potential impact of the SST smoothing on the upper levels, we detect the position of jet following the algorithm of Spensberger et al. (2017). The algorithm detects jets by their axis, the line of maximum winds that separates cyclonic from anticyclonic wind shear. The algorithm requires the wind maximum to be well-defined, but does not impose a strict minimum wind speed (Spensberger et al., 2017). We added this information in subsection 2.2 (lines: 131-144).
For the climatologies (Figure 3) and composites we normalise the occurrence of both SST front lines and jet axis lines to account for the varying area covered by grid cells. We achieve this by showing the average length of SST front line/jet axis line per unit area, hence the resulting unit of length per area. For details on the normalisation, we refer to Spensberger and Spengler (2020). To compile a climatology, we use for both the SST front and the jet axis distributions a unit of length per area, due to its independence of the grid resolution. We added this information in subsection 2.2 (lines: 135-138).

**Figure 4: I find the wording of this caption extremely confusing.**

Following the reviewer's concern, we rephrased the caption of Figure 4.

**Line 185- In my opinion, this hypothesis should be removed as it implicitly assumes an SST influence on the jet, which is meant to be a topic of exploration here. Similar comments apply to Line 188 and Line 288.**

We thank the reviewer for this comment. Considering the reviewers opinion, we removed the hypothesis in line 185.

However, we kept the hypotheses for lines 188 and 288, as previous studies confirmed both the significance of a strong SST gradient for anchoring the storm track (e.g., Nakamura et al., 2004,2008; Sampe et al., 2010), as well as the relationship between moisture and temperature (here between humidity and SST) through the Clausius-Clapeyron relation.

**(M): Line 200 - I cannot understand the logic behind not just looking at all cyclones.**
**Line 223 - you make the bold claim that cyclone intensification is only weakly modified by SST gradient, but what about all those that aren't undergoing maximum intensification right there?**

The Gulf Stream and Kuroshio regions are the areas where the major SST fronts are located and where the maximum SST differences due to smoothing occur. We wanted to relate the intensification of cyclones with the changes in absolute SSTs and/or SST gradient and therefore did not include cyclones with a maximum intensification further away towards the central and eastern parts of the two basins, as the intensification of these cyclones could not be directly associated with these changes in SST.

**(M): Line 258 - This is a main point of the paper and one I don't feel comfortable with. Your claim is that the SST gradient is not particularly important for the intensification of individual cyclones. Whilst I may agree with this general sentiment (see my first major comment), I do not think the analysis presented here can make that conclusion.**
**1) As mentioned previously, you are looking at a subset only.**

We thank the reviewer for sharing these thoughts with us. First of all, please let us clarify that we only used a subset of cyclones (C1,2,3) to investigate the possible changes on cyclone intensification for cyclones belonging to the different categories, for the reasons mentioned in the previous response. For the remainder and main part of the manuscript, however, we use all cyclones. For our previous climatological analysis, we used all cyclones in the respective regions (irrespective of track and location of maximum intensification). For the revised manuscript, we actually use all cyclones in our database. We pointed out this difference between the parts of the paper more clearly (e.g. lines: 95-99, 166-171, 439-440).

**2) There are many differences between your two experiments, not just the SST gradient. The absolute SST changes also, and I strongly suspect that the variability in the Gulf Stream path length/separation from coast changes too, each having been shown to significantly affect the interaction. I don't see anything up to this point that allows you to definitively link it to the SST gradient. You mention in Line 40 that these factors need to be teased apart, but I am not convinced that has been done here simply by defining C1, C2 and C3 - these classifications are also ill-defined in the SMTH experiment.**

We fully agree with the reviewer regarding the change of both the absolute SSTs and the SST gradient after the SST smoothing, as well as to their proven importance on cyclone intensification. Thus, this issue is frequently discussed in our paper (e.g., lines: 24-41, 186-191, 267-274, 357-362, 381-383, 435-438, 464-466) as well as in our ERA-Interim-based analyses for the same regions (Tsopouridis et al. 2020a,b). One of our conclusions is that both the absolute SST and the SST gradient play a role for the evolution of cyclone characteristics. For instance, the increase (decrease) of the absolute SST results in an increase (decrease) of the upward surface heat fluxes and specific humidity, whereas a weak SST gradient in the SMTHG experiment is found to be associated with a considerable reduction in precipitation (in line with Kuwano-Yoshida et al., 2010b), compared to the pronounced SST gradient in the CNTL experiment. Moreover, we document the changes in SST prior and after the

SST smoothing (Figure 6) and how these changes can affect the intensification of the different cyclone categories, that is C1,2,3 (Figure 5), as discussed in subsection 3.3. Nevertheless, we acknowledge the reviewer's concern (based on Line 40) that the two factors have not been separated as thoroughly as in Tsopouridis et al. (2020a,b), and thus rephrased our sentence in the Introduction (please refer to Section Introduction, lines: 40-41).

Regarding the "variability in the Gulf Stream path length/separation from coast", the heavy smoothing in the SMTHG experiment implies that there is actually not even a "Gulf Stream" anymore, not only that the length has changed.

**Line 266- as I alluded to earlier, it does not seem appropriate to use two different subsets of cyclones for each of these sections (individual vs. mean state) if you are going to draw comparisons between them. Additionally, given that the typical timescale for thermal air-sea interaction is ~ a day or so, one would expect the GS/KE to impact some storms further downstream from the region. This again raises questions regarding the conclusion that the SST front does not affect individual cyclones, just because the ones reaching maximum intensity in that specific regions are not that affected.**

Regarding the subsets we refer to our response to L258/point 1.

Regarding the debate on local versus remote effects, we appreciate that our previous climatological analysis was not ideal to comment on this issue, but the revised analysis allows us to comment on this issue. Counting all air-sea exchange within different radii of a cyclone centre as "local", we now show that "non-local" air-sea exchange dominates the climatology. This finding is in line with previous studies that tried to clarify the influence of surface heat fluxes on midlatitude cyclone development and untie their local and remote effects (e.g., Nuss and Anthes 1987; Kuo and Reed 1988; Haualand and Spengler 2020; Bui and Spengler 2021). We thus believe that SSTs and SST fronts are mainly important for climatologically setting the environment in which cyclones evolve, though each individual cyclone is not significantly affected directly by changes in the SST.

**(M): Section 3.4. Line 294 -Parfitt and Czaja (2016) have shown that the top 30% of climatological latent heat fluxes in the Gulf Stream region are associated with cyclones that have just passed over the region. It is continuously said here "when no cyclones are present in the respective region", however that simply means the cyclone center is not directly above the region - this does not mean a cyclone's passing isn't responsible for the trailing cold air that maximises the fluxes. This is relevant for consistency with Section 3.4.2 later, where it is shown cyclone precipitation does change significantly. A lot of moisture availability, that is ultimately taken up in say the warm conveyor belt of any given cyclone, in these regions can come from the strong latent heat fluxes**
**behind the cold front of a system that passed ahead of it. This comment is also relevant for your statement "for specific humidity, cyclones account only for a small part of the climatological differences" in Line 359, and for your statement from Line 379-383. In particular, I do not agree with Line 381-383. This comment feeds back to the previous one, where I do not think conclusions about the SST influence on the atmosphere through cyclones can be drawn simply by comparing whether a cyclone is in that specific region or not.**

We acknowledge the reviewer's concerns and agree that the attribution to only cyclone centres in the box might have been too restrictive. Please refer to our response in the first comment on the additional analysis we now present in the revised version of the manuscript where we include all cyclones in the respective ocean basin.

**Figure 8: The colors on Figure 8 (b-h) I would recommend changing as it is hard to see any difference.**

We adjusted the colours, so that the differences are more evident in the revised version of the manuscript.

**Section 3.4.4 - I would recommend mentioning Lee et al. (2018) here also.**

We thank the reviewer for his suggestion and making us aware of this study. Nevertheless, we decided instead to mention the study of Lee and Kim 2003, as it better reflects our findings and discussion in subsection 3.4.4.

**Lastly, I noticed a number of spelling/grammatical errors, I suggest a thorough check.**

We thank the reviewer for making us aware of these errors/typos, which we adjusted accordingly in the revised version of the manuscript.

**References**

Bishop, S. P., Small, R. J., Bryan, F. O., & Tomas, R. A. (2017). Scale dependence of midlatitude air–sea interaction. Journal of Climate, 30(20), 8207-8221.

Bui, H., & Spengler, T. (2021). On the Influence of Sea Surface Temperature distributions on the Development of Extratropical Cyclones. Journal of the Atmospheric Sciences.

Catto, J. L., Jakob, C., Berry, G., & Nicholls, N. (2012). Relating global precipitation to atmospheric fronts. Geophysical Research Letters, 39(10).

Colucci, S. J., & Davenport, J. C. (1987). Rapid surface anticyclogenesis: Synoptic climatology and attendant large-scale circulation changes. Monthly weather review, 115(4), 822-836.

Dallavalle, J. P., & Bosart, L. F. (1975). A synoptic investigation of anticyclogenesis accompanying North American polar air outbreaks. Monthly Weather Review, 103(11), 941-957.

Haualand, K. F., & Spengler, T. (2020). Direct and Indirect Effects of Surface Fluxes on Moist Baroclinic Development in an Idealized Framework. Journal of the Atmospheric Sciences, 77(9), 3211-3225.

Hanley, J., & Caballero, R. (2012). Objective identification and tracking of multicentre cyclones in the ERA-Interim reanalysis dataset. Quarterly Journal of the Royal Meteorological Society, 138(664), 612-625.

Kuo, Y. H., & Reed, R. J. (1988). Numerical simulation of an explosively deepening cyclone in the eastern Pacific. Monthly Weather Review, 116(10), 2081-2105.

Kuwano-Yoshida, A., Minobe, S., & Xie, S. P. (2010). Precipitation response to the Gulf Stream in an atmospheric GCM. Journal of Climate, 23(13), 3676-3698.

Kuwano-Yoshida, A., & Minobe, S. (2017). Storm-track response to SST fronts in the northwestern Pacific region in an AGCM. Journal of Climate, 30(3), 1081-1102.

Lee, S., & Kim, H. K. (2003). The dynamical relationship between subtropical and eddy-driven jets. Journal of the atmospheric sciences, 60(12), 1490-1503.

Masunaga, R., Nakamura, H., Miyasaka, T., Nishii, K., & Tanimoto, Y. (2015). Separation of climatological imprints of the Kuroshio Extension and Oyashio fronts on the wintertime atmospheric boundary layer: Their sensitivity to SST resolution prescribed for atmospheric reanalysis. Journal of Climate, 28(5), 1764-1787.

Masunaga, R., Nakamura, H., Taguchi, B., & Miyasaka, T. (2020a). Processes Shaping the Frontal-Scale Time-Mean Surface Wind Convergence Patterns around the Kuroshio Extension in Winter. Journal of Climate, 33(1), 3-25.

Masunaga, R., Nakamura, H., Taguchi, B., & Miyasaka, T. (2020b). Processes shaping the frontal-scale time-mean surface wind convergence patterns around the Gulf Stream and Agulhas Return Current in winter. Journal of Climate, 33(21), 9083-9101.

Murray, R. J., & Simmonds, I. (1991). A numerical scheme for tracking cyclone centres from digital data. Part I: Development and operation of the scheme. Aust. Meteor. Mag, 39(3), 155-166.

Nakamura, H., Sampe, T., Tanimoto, Y., & Shimpo, A. (2004). Observed associations among storm tracks, jet streams and midlatitude oceanic fronts. Earth's Climate: The Ocean–Atmosphere Interaction, Geophys. Monogr, 147, 329-345.

Nakamura, H., Sampe, T., Goto, A., Ohfuchi, W., & Xie, S. P. (2008). On the importance of midlatitude oceanic frontal zones for the mean state and dominant variability in the tropospheric circulation. Geophysical Research Letters, 35(15).

Nakamura, T., Yamazaki, K., Iwamoto, K., Honda, M., Miyoshi, Y., Ogawa, Y., ... & Ukita, J. (2016). The stratospheric pathway for Arctic impacts on midlatitude climate. Geophysical Research Letters, 43(7), 3494-3501.

Neu, U., Akperov, M. G., Bellenbaum, N., Benestad, R., Blender, R., Caballero, R., ... & Wernli, H. (2013). IMILAST: A community effort to intercompare extratropical cyclone detection and tracking algorithms. Bulletin of the American Meteorological Society, 94(4), 529-547.

Nuss, W. A., & Anthes, R. A. (1987). A numerical investigation of low-level processes in rapid cyclogenesis. Monthly Weather Review, 115(11), 2728-2743.

O'Neill, L. W., Haack, T., Chelton, D. B., & Skyllingstad, E. (2017). The Gulf Stream convergence zone in the time-mean winds. Journal of the Atmospheric Sciences, 74(7), 2383-2412.

Parfitt, R., & Czaja, A. (2016). On the contribution of synoptic transients to the mean atmospheric state in the Gulf Stream region. Quarterly Journal of the Royal Meteorological Society, 142(696), 1554-1561.

Parfitt, R., & Seo, H. (2018). A New Framework for Near Surface Wind Convergence Over the Kuroshio Extension and Gulf Stream in Wintertime: The Role of Atmospheric Fronts. Geophysical Research Letters, 45(18), 9909-9918

Rudeva, I., & Gulev, S. K. (2011). Composite analysis of North Atlantic extratropical cyclones in NCEP–NCAR reanalysis data. Monthly weather review, 139(5), 1419-1446.

Sampe, T., Nakamura, H., Goto, A., & Ohfuchi, W. (2010). Significance of a midlatitude SST frontal zone in the formation of a storm track and an eddy-driven westerly jet. Journal of Climate, 23(7), 1793-1814.

Tsopouridis, L., Spensberger, C., & Spengler, T. (2020). Characteristics of cyclones following different pathways in the Gulf Stream region. Quarterly Journal of the Royal Meteorological Society.

Tsopouridis, L., Spensberger, C., & Spengler, T. (2020). Cyclone intensification in the Kuroshio region and its relation to the sea surface temperature front and upper-level forcing. Quarterly Journal of the Royal Meteorological Society.

Wernli, H., & Schwierz, C. (2006). Surface cyclones in the ERA-40 dataset (1958–2001). Part I: Novel identification method and global climatology. Journal of the atmospheric sciences, 63(10), 2486-2507.

---

## Referee Report (RR1)

The Gulf Stream and Kuroshio SST fronts affect the winter climatology primarily in the absence of cyclones
wcd-2020-50

This paper aims to determine the influence of SST gradients on extratropical cyclone intensification and changes to the environment due to cyclones in a world in which SST gradients are smoothed. The authors address these aims using 2 methods. Firstly, they artificially smooth SST gradients in the Gulf Stream and Kuroshio regions and examine cyclone development differences. They demonstrate that smoothing the SST gradients in both the Gulf Stream and Kuroshio regions does not result in significant differences to cyclone intensification but that the total number of cyclones reduces. Secondly, they partition the atmosphere into regions that are inside/outside the region directly influenced by cyclones and compare the differences in the control and smoothed SST gradient simulations. This demonstrates that the differences between the control and smoothed SST gradients simulations are larger when there are no cyclones present. While I understand the rationale behind the first aim, I do not follow the motivation for the second aim or totally agree with the conclusion that they reach. Therefore I think this paper requires revising before it is suitable for publication in WCD.

General comments
1. It is not clear to me what the motivation for this study is. Why perform the smoothed SST simulations in the first place? Are we expecting the SST gradients in the Gulf Stream and Kuroshio current to change in the future? Are the authors trying to say something about the response of the climate in coarse resolution models with low ocean resolution?
2. The conclusion from the second aim is ambiguous. The results show that the influence of cyclones on environmental changes due to smoothing the SST gradients is small. Does this mean that cyclones do not influence the environment in either simulation, or that their influence is large in both simulations but does not depend on the underlying SST gradients?
3. Furthermore, the authors conclude that 'cyclones play only a secondary role in explaining the mean state differences between the smoothed and realistic SST simulations'. To what extent are the mean state differences because there are fewer cyclones, i.e., it is the absence of cyclones in the smoothed SST gradient simulations that results in the large differences. If this is the case, then you could say that changes in the storm track position and a reduction in the number of cyclones play the dominant role in explaining the mean state differences. Perhaps this perspective is what the authors are referring to with their 'direct' and 'indirect' terminology? If so, this needs to be clarified.
4. I did not understand the title. What climatology are they referring to?
5. It has been shown by Vanniere et al. (2017) and recently by Marcheggiani and Ambaum (2020) that cyclones tend to destroy the low-level temperature gradient within the cold sector due to a strong air-sea heat fluxes, but that it is restored within a few days following the cyclone passage. Could the authors comment on whether their spatially defined results for cyclone and non-cyclone environments are consistent with this temporal analysis.

---

## Author Response (AR2)

**Response to Referee 1 (Irina Rudeva)**

*My major concern is that the paper does not offer explanation as to why SST fronts seem to have little influence on cyclone characteristics. I would like to see if not a detailed analysis on this, but at least discussion.*

We understand the reviewers wish and added several discussion items as further outlined below.

*Despite the absence of an SST front in the Atlantic, the SST gradient over the black box remains strong compared to the rest of the ocean. Thus, even in the absence of a sharp SST front, this gradient may remain efficient in creating baroclinic instability. Jacobs et al. (2005) offered a very simple prestorm baroclinic index PSBI = (TG - Tland)/d, where d is the distance to the Gulf Stream from the coastline. The results shown in the paper make me think that land-sea contrasts might be significantly larger than the SST difference between CNTRL and SMTH experiments. I suggest calculating a similar index for cyclones in C1-C3 categories to check this (along which line this gradient should be calculated is an open question, e.g., along the same latitude as the cyclone centre at max intensity).*

We thank the reviewer for raising this issue. Tsopouridis et al., 2020a,b (https://doi.org/10.1002/qj.3924 and https://doi.org/10.1002/qj.3929 ) related the intensification of individual cyclones to the land-sea contrast (in the Gulf Stream region) and to the upper-level forcing (particularly in the Kuroshio region), while they found the SST front to only play a secondary role for the intensification of cyclones in the two regions. We included this discussion in the revised version of the manuscript (lines: 256-258, 300-301). Moreover, to estimate the relative role of the land-sea contrast and the SST front, we conducted a composite analysis for cyclones in the three categories (C1-C3) for the North Atlantic, where both the land-sea contrast and the SST gradient are more pronounced, but only the latter was considerably changed in the SMTH experiment. We included this additional analysis in the supplement (Figures S11, S12) and discuss the results in lines: 258-266. Our results indicate that despite the significantly reduced SST gradient in the SMTHG experiment, the low-level baroclinicity (based on the temperature gradient at 850 hPa) remains largely unchanged, indicating the dominant role of the land-sea contrast to enhance low-level baroclinicity and hence cyclone intensification compared to the SST front, which is fully in line with the reviewer's thoughts.

*The other important question is why SST fronts play a larger role in the absence of cyclones. While land-sea contrasts, discussed above, can be particularly important in case of already formed cyclones, sharp SST fronts may play a large role during less perturbed flows, creating thermal wind and affecting the circulation downstream.*

We are not sure what the reviewer refers to as "less perturbed flows". If one would argue that the flow is less perturbed in the absence of cyclones, the argument of the reviewer actually aligns with our results and interpretation, i.e., that the non-cyclone time steps and thus less perturbed situations play a more crucial role in terms of how the SST front imprints itself on the mean state.

*Title: climatology of what? I'd replace 'climatology' with 'atmospheric flow'. The title still reads like a media headline rather than a name of a scientific paper, but I feel that this is the impression the authors wanted to make.*

We thank the author for this comment. With this title, we wanted to highlight the main finding of our study and did not intend to make a "media headline". Explicitly stating "atmospheric flow" instead of climatology would not be correct, as we also address variables, such as precipitation and surface fluxes, that do not describe the atmospheric flow.

Am I right that cyclones selected for the analysis spent 3 6-hr steps and reached their max intensity within the black boxes? All other cyclones fell into category 'absence of cyclones'? l 537-8, 539: This is only true for cyclones that reach their max intensity over the Gulf Stream/Kuroshio, if I got it right. Cyclones downstream of the currents may still be modified by changing SST gradients (and baroclinicity) and play a role in forming those anomalies.

As indicated in the first revised version of the manuscript, we consider cyclone intensification not intensity (e.g., lines: 160, 228-229, 243-246, 247-255, 258-266, 2659-272, 275, 287, 300-301, 452-455, 491-493). Further, for the decomposition of cyclones, we outline in the manuscript that we consider all cyclones propagating in the North Atlantic and the North Pacific "irrespective of the direction of cyclone propagation and location of maximum intensification" (see lines: 165-171, 439-440 in the previous version of the manuscript). The only limitation is that we consider cyclone tracks with at least five 6-hour time steps, following a commonly used technique (Neu et al. 2013). We thus feel that the data selection is sufficiently clear in the current version of the manuscript.

Why max wind speed is significantly displaced equatorward in the absence of cyclones but not when they are present? Is it because the jet is already displaced southward in both CNTL and SMTH simulations when a cyclone is present?

In the Atlantic, the displacement is present for both when cyclones are present and absent. The different intensity of the displacement pattern mainly reflects the fact that the region features more time steps with no cyclone present. See also the respective climatology in the supplement (Fig. S13). For the Pacific, the story is similar, though with some differences between the cyclone/no-cyclone patterns in the westernmost part of the Pacific, which we do not have a good hypothesis for. We included a brief discussion about this in the manuscript, though cannot provide a conclusive answer explaining these discrepancies (see lines 434-440).

l. 245; Could slowing of the jet over the Gulf of Alaska along with increasing cyclone density be related to higher cyclolyse in that region?

We agree with the reviewer that a slowing of the jet over the Gulf of Alaska along with increasing cyclone density can indeed be related to higher cyclolysis in that region. However, a further analysis of this geographic region would be beyond the scope of this paper and we thus decided to not include this discussion in the revised version of the manuscript.

Minor comments:

l.536: decrease in SST contrast

We thank the reviewer for pointing this out. We filled the missing word in the revised version.

l. 537: Did you mean SST gradient, not just SST?

Yes, we meant SST gradient and adjusted the text accordingly.

Fig. 4: I would suggest to show statistically significant differences. It is also interesting to see the trajectories of cyclones in supplements.

Following the reviewer's request, we revised Figure 4 to show only statistically significant differences. We also present the trajectories of all cyclones for the two regions in the supplement for all the experiments as well as the trajectories of cyclones belonging in the three sub-categories (C1,2,3), as suggested by the reviewer (Figure S9,S10). Finally, following the editor's suggestion we included the frequencies for the "within/outside" cyclone area analysis (Figure S13).

No, the SSTs are averaged over a 400km radius. Sensitivity tests using different radii (e.g., 200km) were conducted and yielded similar results. The 750km radius was used only for the second part of our analysis ("presence/absence of cyclones") following the reviewers' comments during the first round of reviews (see lines 174-180, 333-336, 476-478).

Fig.7-10: It is said that differences are statistically significant at 95% level. Is it true for all grid points even with differences close to zero? There are white areas in panels b and f, but not in c and g.

We acknowledge that the caption was confusing, giving the impression that a statistical test was also conducted for the "within/outside" analysis (panels c,d and g.h) for figures 7-10, which was not the case. However, instead of editing the caption, we decided to perform statistical tests for panels c,d & g,h and present the respective results in figures 7-10.

**Response to anonymous referee 2**

- Please describe these papers correctly. The Parfitt and Seo (2018) paper indicates the importance of atmospheric fronts and the baroclinic waveguide (i.e. the combination of both cyclones and anti-cyclones), not extratropical cyclones specifically. This distinction is important. Also, many frontal detection algorithms detect atmospheric fronts in the Gulf Stream / Kuroshio regions without an "associated" extra tropical cyclone, especially with weaker fronts, as in Masunaga et al. (2020a, b). Simply including a larger cyclone radius does not address the issue of fronts discussed in previous work.

We thank the reviewer for this comment and agree that the citation should have been more on point and we thus rephrased this sentence (see lines: 80-81). Further, while we agree with Parfitt and Seo (2018) about the role of the baroclinic waveguide for the propagation of Rossby waves and thus cyclones and anticyclones, such a wave guide is usually constituted by neither cyclones nor anticyclones, but by the jet (c.f. review in Martius et al. 2010, https://doi.org/10.1175/2009JAS2995.1). As we discuss the impact on the position and intensity of the jet, we thereby also address the waveguide (e.g., see lines: 43-49, 138-141, 208-219, 273-275, 425-428, 441-443), similar to TSS20a,b.

Regarding the role of fronts, we would like to refer the reviewer to a recent publication (Reeder et al. 2021, https://doi.org/10.1175/JAS-D-20-0118.1) which complements the analysis in the present manuscript by focussing on fronts rather than cyclones. Given that Reeder et al. (2021) document that the SST front imprints itself mainly in the absence of atmospheric fronts, it is a timely and pertinent question to ask whether the same might be true for cyclones. We now explicitly refer to this study and put our work in context (see lines: 84-86).

Regarding the association of variables to cyclones by defining a cut-off radius, we follow a standard analysis method (c.f. Rudeva and Gulev 2011; and Catto and Pfahl 2013, https://doi.org/10.1002/jgrd.50852). Catto and Pfahl (2013) associate precipitation with fronts by the same technique. With a maximum radius of 1000 km we make sure to include all fronts associated with the cyclone into our analysis, such that our analysis appears well-suited to address the questions we set out to answer.

The Bishop et al. (2017) paper is relevant as it shows a clear shift to ocean-driven variability. Numerous other papers show you need eddy-resolving SSTs to fully understand this variability and its influence on the atmosphere. e.g. Liu et al. (2021) - regardless of prescribed SSTs or not. Liu, X., Ma, X., Chang, P., Jia, Y., Fu, D., Xu, G., ... & Patricola, C. M. (2021). Ocean fronts and eddies force atmospheric rivers and heavy precipitation in western North America. Nature communications, 12(1), 1-10.

-The above comment leads to the authors response regarding the lack of ocean eddy-resolving resolution in their simulations. I have two issues with their response. 1) It is not a valid argument to say that because the simulations and datasets have been used for studies previously, that they are suitable to use now. One would no longer solely use the CMIP2 models or ERA-40 to model climate variability, despite them being used extensively in the past. As mentioned above, many papers show lack of eddy-resolution in the ocean results in incorrect atmospheric responses. 2) In my opinion, papers like the one mentioned above clearly demonstrate the authors assertion that "it seems implausible that ocean eddies will fundamentally change… 1st order effect on synoptic-scale systems" is incorrect.

We acknowledge the reviewers point about the potential impact of mesoscale ocean eddies on air-sea heat fluxes and are aware that this has been documented in previous studies, including the ones

the reviewer mentions. While some recent studies focus on the impact of mesoscale eddies, there is a large body of literature on how larger scale SST gradients along the two of the major western boundary currents impact the atmosphere. We aim to investigate this impact and ask how and if the SST gradient *directly affects* the evolution of individual cyclones. Or, alternatively, if the SST gradient predominantly affects the environment in which the synoptic systems occur, and thus *indirectly affects* individual cyclones. As argued in our manuscript, our findings indicate that the *indirect impact* on cyclone development is dominating the *direct impact*, because, as shown in our results, the SST gradient imprints itself on the atmosphere mainly in the absence of cyclones. Given the insensitivity of individual cyclones to such drastic changes in the SST gradient as considered here, we find it very plausible that also the *direct impact* of mesoscale eddies in the ocean is small. Nevertheless, similar to the large-scale SST gradient, we believe that ocean eddies can play a role in shaping the atmospheric mean state, potentially also mainly through interactions in the absence of cyclones. However, this investigation is beyond the scope of our study. We provided more context for our studies vis-à-vis the role of ocean eddies in our revised introduction (see lines: 60-62).

We hence believe that the usage of AFES data is justified, as we only focus on 1st-order effects on synoptic scale systems and on the potential direct effect of the SST fronts on the evolution of cyclones. The same dataset has been extensively used in the past in numerous respective analyses, as explicitly described in the manuscript (e.g., lines: 109,115,308). We acknowledge the reviewer's concern on the coarser resolution of the AFES model, but after comparing the AFES climatology with the ERA-Interim dataset we found the air-sea heat exchange, as well as the storm track to be reasonably represented. Apart from this, we previously tested the results for the two time periods of the ERA-Interim reanalysis (1979-2001 and 2002-2016) to test the possible impact of the resolution change in SST, with overall no significant change. We believe that just using a more recent dataset would not constitute a more novel approach to the problem that we are tackling in our manuscript.

We argue consistently within our findings and our arguments are backed up by the analysis that we present. Thus, we disagree with the reviewer that we make incorrect statements. Our statements are based on our analysis, which we believe is sound. If the reviewer disagrees with our analysis, the reviewer should clearly outline flaws in our methodology. We regard stating an opinion that our claims are incorrect, despite them being firmly based on our analysis and presented results, insufficient as a criticism. Time will show if future analyses and methodology will support our findings or refute them.

- I still do not believe that you can draw strong comparisons between the Gulf Stream and Kuroshio if the definitions used in each basin are different. I understand the magnitude of the SST gradients in the two basins are different, but I cannot accept the argument that this means it is impossible to come up with an SST front metric that fits both. This is also related to a comment I have on the SST gradient definition between CNTL and SMTHG/K (see below).

In general, it would be best to use a metric that "fits both" basins, rather than conducting sensitivity tests for each region to identify suitable thresholds. However, the Gulf Stream and Kuroshio regions feature certain differences, such as a distinct upper-level wave field, a different structure and intensity of the upper-level jet, and a significantly greater SST gradient in the Atlantic compared to the Pacific (e.g., Nakamura et al., 2004; Tsopouridis et al., 2020b). Thus, choosing the same threshold to detect the SST front in the two regions, would indicate that we set aside one of the most important differences between the two basins, which we argue would lead to questionable results in both regions (underestimation/overestimation of SST fronts, respectively). We thus strongly believe that a different threshold is necessary to accommodate for the different natures of the boundary currents and SST fronts together with the overall different characteristics in the two regions. These arguments are also clearly presented in our manuscript (Lines: 131-137).

- Related to my comment earlier, I still have a serious concern with using the SST front definition in the CNTL experiment to define the location in the SMTHG/K. The fact that no SST front is identified in the SMTHG/K with the current SST front definition simply tells me the definition needs to be altered. I suggest the authors review the numerous papers that have defined the Gulf Stream / Kuroshio location or the associated SST gradient in data as coarse as that used here. I have to say that this issue makes it extremely difficult for me to be confident in any of the CNTL vs. SMTHG/K comparisons. Especially as this makes the definitions for C1, C2, and C3 have little meaning.

We thank the reviewer for this comment. "Reviewing the numerous papers that have defined the Gulf Stream/ Kuroshio location or the associated SST gradient in data as coarse as that used here" was our first priority and we indeed referred to the respective studies several times, particularly to the ones from Kuwano-Yoshida and Minobe (2017) and Kuwano-Yoshida et al. (2010), in which the same data were used (e.g., in lines: 21-22, 29-47, 52-54, 64-72, 111-117, 285-294, 335-337, 343-344 in the previous version of the manuscript). The fact that no SST front is identifiable in SMTHG/SMTHK is not a result of "the current SST front definition", but arises from the extent to which the SSTs are smoothed. In a region where the SST gradient is constant over several hundred kilometres, the concept of SST front simply does not longer apply, irrespective of the method chosen to identify fronts. We would understand the reviewer's concerns if a considerably weaker SST gradient would have resulted in a significantly reduced number of cyclones or generally different climatology of cyclones in the region. However, as presented in Figures S1,S2 in the revised version of the manuscript, an almost equal number of cyclones of C1,2,3 propagate roughly in the same region in the experiments with the smoothed SST. While the approach for our C1,2,3 analysis comparing CNTL and the smooth experiments might not be fully optimal, we argue that it is a valuable compromise for the comparison that we present.

- The authors did not make an attempt to actually show how sensitive their results (ocean influence on atmosphere) in this paper are to cyclone detection. I would be surprised if the results were not sensitive to this, especially if the differences in "shallow and weak systems" are noticeable between algorithms as the authors mention.

We thank the reviewer for letting us share our thoughts on this. The different algorithms used for cyclone detection provide different results, first and foremost due to the different atmospheric fields used for defining a cyclone, with mean sea level pressure (MSLP) or lower tropospheric vorticity being the basic tracking metrics (e.g., Sinclair 1994; Hodges et al. 2003; Rudeva and Gulev 2007; Ulbrich et al. 2009). As stated in Neu et al., 2013 "there is no accepted universal definition of what a cyclone is or where its exact position is". However, for the analysis in Tsopouridis et al., 2020a, we thoroughly tested the sensitivity of the results and found that even when using the same metric to define a cyclone, the results are sensitive to the choice of the several parameters, which is also evident from Figure 1 in Neu et al., 2013. We added this information in the manuscript (lines: 150-151). Unfortunately, the great majority of the studies do not provide a detailed namelist of the chosen parameters. In Tsopouridis et al. (2020a), we publish the values of the parameters for the detection and tracking namelists. Overall, we want to highlight, what is underlined in Neu et al., 2013 (to our knowledge the most complete study on cyclone detection and tracking algorithms) and with which we fully agree: "since there is no universal agreement upon cyclone definition, we cannot "judge" the algorithms or say that a specific one delivers "incorrect" results. They are all "right" in some sense."

- Regarding the logic behind only looking at cyclones undergoing maximum intensification right there, I understand the authors point that the maximum intensification of cyclones away from the SST front could not be directly associated with changes in SST. But that doesn't at all mean that they can't be. And in fact, it is fairly well understood that many ocean induced impacts are related to maximum

intensification elsewhere (in which case you are not considering the whole set of scenarios). For example, changing the absolute SST will alter the magnitude and location of the evaporation input into the atmosphere. One can expect scenarios where the maximum impact of anomalous latent heating within the cyclone will occur downstream once moisture has risen within the system. This is also relevant for the authors response to a later comment (" we thus believe that SSTs … are mainly important for climatologically setting the environment in which cyclones evolve, though each individual cyclone is not significantly affected directly " ). What about an SST change that moistens the atmosphere prior to the cyclone arrival, but then the cyclone intensifies maximally downstream?

Our results do not contradict the arguments of the reviewer, in fact, we feel that the reviewer paraphrases our findings. Our argument is that the direct impact of the SST front on cyclone development is rather negligible, while the SST changes certainly have an imprint on the climatological setting that will influence cyclone development. In particular, we actually present that there are significant changes in evaporation when SSTs are smoothed. However, we also show that the direct impact on cyclones appears to be small, whereas the indirect effect can contribute to alterations in cyclone development, which is consistent with other recent findings (Bui and Spengler, 2021; Haualand and Spengler, 2020) (see lines: 89, 302-303). We further clarified this line of argument in the manuscript (see lines 491-494).

- For many of the reasons stated above, I still do not think the results in this manuscript can be used to claim that ' SST fronts only have a minor impact on the characteristics and intensification of individual cyclones'. I would like to also point out that just because differences between CNTL and SMTHG/K are greater in some variables outside a cyclone radius, does not mean that SST fronts have a minor impact on individual cyclones.

We would like to refer to our response to the previous comment. We agree that changes in the SST can have an impact on cyclones, though argue that this influence is indirect, where the changed SST leads to a different baroclinicity and moisture availability that will then influence cyclone development. These arguments are in line with other recent findings clarifying the direct and indirect effects of surface fluxes on cyclone development (Haualand and Spengler, 2020; Bui and Spengler, 2021, lines: 89, 302-303, 491-494).

---

## Author Response (AR3)

**Response to Referee #1 (Irina Rudeva)**

We thank Irina Rudeva for another round of constructive feedback on our manuscript and are pleased to read that we were able to address most of her concerns.

Regarding the reviewer's concern about the "provoking title", we agree that the wording is probably too strict. However, our study is not only about the "cyclone response", as we explicitly analyze how the climatological fields are composed conditioned on times with cyclones present and absent. It is correct that we cannot provide a conclusive argument for the observed differences, though climatological studies often have a more descriptive character. We agree that the verb "affect" might be too strong as it implies a causality that we did not prove. Furthermore, it is not strictly the climatology that is not affected by cyclones, but the response to changes in the intensity of the SST gradient that is not affected by the presence of cyclone. We thus thank the reviewer for her continued criticism of the title that now made us reconsider its formulation. We changed the title to "Smoother versus sharper Gulf Stream and Kuroshio SST fronts: Effects on cyclones and climatology" and hope that the new title addresses the reviewer's concerns. It now consistently reflects the presented analysis in our manuscript.

Response to minor comments:

l.138: in line
We thank the reviewer for indicating this typo.

l.221: perhaps, 'elongated' is better than 'distributed''
We agree with the reviewer and replaced "distributed" with "elongated".

l.437: In my opinion, fig. S13 shows only marginal increase in the number of cyclones. Even if the difference is significant, it is not clear to me how it explains the equatorward shift of the jet.
We agree that the increase in number might be small and reformulated the argument to reflect that the difference can be only partially attributed to the differences in cyclone presence. We have already previously indicated that we do not have a causal explanation for the jet shift and the paragraph in question also does not attempt to causally explain this shift, but to merely attribute those parts of the signal that might be interpretable based on the data at hand. To properly attribute the jet shift additional numerical simulations and a more extended analysis would be necessary, which is beyond the study at hand.

L.514-515: I find this statement on the direct influence of the SST front to be very strong. As said in Reeder at al. 'strong localised diabatic frontogenesis, which is amplified by adiabatic frontogenesis, can result in a front, which is consistent with atmospheric fronts in the region being most frequently located along the SST front' - I fully agree with this statement and new plots (S12-13) make me think that a very weak effect on cyclones my be due to characteristics of the model.
We, of course, agree with Reeder et al. (2021), though struggle to contextualize the reviewer's comment about the "weak effect on cyclones [being] due to [the] characteristics of the

model", as we do not understand what "characteristics" the reviewer refers to. In fact, all the lines the reviewer refers to in the comment above only directly relate to the study by Reeder et al. (2021) only dealing with fronts without a direct reference to cyclones. We thus propose to leave the paragraph as it is.

Also, in the phrase 'when no atmospheric fronts are present' replace 'fronts' with 'cyclones', as this paper does not explore atmospheric fronts
We refer to the study by Reeder et al. (2021) in this statement, which explicitly analyzes fronts. Therefore, it is correct to keep 'fronts' in the sentence, as Reeder et al. (2021) did not analyze cyclones. We however rephrased the sentence to avoid the impression that we also considered fronts in the present manuscript.

Fig. S12: It is hard to compare S11 and S12 by eye. Would be good if fig. S12 showed SMTH-CNTRL.
We used these figures to argue that despite the change in the SST gradient between the experiments (CNTL & SMTHG) the low-level baroclinicity (T850, with purple contours) remains relatively unchanged. We explored the suggested possibility and concluded that a difference plot is not so well suited in this case, as the differences at T850 is affected by both changes in location as well as amplitude. Instead of a difference plot, we now include the delta in temperature across the domain, similar to Tsopouridis et al. (2021b), where we studied the ERA-Interim in the Kuroshio region. The presented arguments hold, except for C2, though these are the cyclones propagating away from both the SST gradient and the continent.

**Response to Referee #2**

We thank the reviewer for the constructive comments on our manuscript.

Response to the referee's general comments:

*It is not clear to me what the motivation for this study is. Why perform the smoothed SST simulations in the first place? Are we expecting the SST gradients in the Gulf Stream and Kuroshio current to change in the future? Are the authors trying to say something about the response of the climate in coarse resolution models with low ocean resolution?*
The main motivation is to understand the impact of the intensity of the SST gradient along these western boundary currents on the cyclones developing in these regions as well as how changes in cyclone behavior feeds back on the detected climatological differences. This motivation was clearly stated at the end of the first paragraph in our introduction. We nevertheless rephrased this sentence for further clarity and added a recapitulation of this motivation in the revised last paragraph of our introduction.

We do not claim that the investigated changes in SST are realistic. Similar to previous work with a similar approach, we aim to attain a more mechanistic understanding of cyclone response and their feedback on the climate in dependence on the underlying SST.

*The conclusion from the second aim is ambiguous. The results show that the influence of cyclones on environmental changes due to smoothing the SST gradients is small. Does this mean that cyclones do not influence the environment in either simulation, or that their influence is large in both simulations but does not depend on the underlying SST gradients?*
Thank you for raising this question. We also analyzed the atmospheric fields conditioned on cyclone presence or absence for different variables. For surface fluxes, cyclone and no-cyclone time steps contribute more or less equally to the climatological fluxes, whereas for precipitation, time steps with cyclones present tend to contribute more than when no cyclones are present. However, the main question the manuscript tries to address is how changes in the SST gradient imprint themselves on the climatology and how much these changes can be attributed to changes in cyclone characteristics. For this question, the contribution attributable to the presence of cyclones is small, also for precipitation, despite most of the large-scale precipitation usually occurring in the presence of cyclones. Given that the motivation for this study is to understand the impact of the differences in SSTs on cyclones and the climatological fields, we feel that including a discussion on the general influence of cyclones on the climatology in the Pacific and Atlantic would be beyond this study.

*Furthermore, the authors conclude that 'cyclones play only a secondary role in explaining the mean state differences between the smoothed and realistic SST simulations'. To what extent are the mean state differences because there are fewer cyclones, i.e., it is the absence of cyclones in the smoothed SST gradient simulations that results in the large differences. If this is the case, then you could say that changes in the storm track position and a reduction in the number of cyclones play the dominant role in explaining the mean state differences. Perhaps this perspective is what the authors are referring to with their 'direct' and 'indirect' terminology? If so, this needs to be clarified.*
The total occurrence of cyclones is not altered significantly enough (shown in the supplementary material) to explain the differences. The reviewer's argument was already considered in a previous round of review and we hence already included the cyclone densities in the previous version. We did not find evidence for the change in cyclone densities explaining

significant parts of the signal, which is also what we argue for in our manuscript. Haualand and Spengler (2020) coined the direct and indirect influence of surface fluxes on cyclone development. We now refer to their definition and further clarified this in the manuscript.

I did not understand the title. What climatology are they referring to?
In response to concerns from both reviewers we changed the title to "Smoother versus sharper Gulf Stream and Kuroshio SST fronts: Effects on cyclones and climatology". We hope the revised title addresses the concern of the reviewer and makes it more understandable.

It has been shown by Vanniere et al. (2017) and recently by Marcheggiani and Ambaum (2020) that cyclones tend to destroy the low-level temperature gradient within the cold sector due to a strong air-sea heat fluxes, but that it is restored within a few days following the cyclone passage. Could the authors comment on whether their spatially defined results for cyclone and non-cyclone environments are consistent with this temporal analysis.
This is an interesting point and we agree that our results could be interpreted in a similar way, though their definition of cold sector might often lie outside what we refer to as cyclone area. It has also been shown that cold air outbreaks, which contribute most significantly to the air-sea heat exchange, are sometimes only remotely associated with synoptically developing cyclones. We edited the manuscript to contextualize our findings with these studies.